# Genetic studies of accelerometer-based sleep measures yield new insights into human sleep behaviour

Samuel E. Jones ⓘ et al.[#]

Sleep is an essential human function but its regulation is poorly understood. Using accelerometer data from 85,670 UK Biobank participants, we perform a genome-wide association study of 8 derived sleep traits representing sleep quality, quantity and timing, and validate our findings in 5,819 individuals. We identify 47 genetic associations at $P < 5 \times 10^{-8}$, of which 20 reach a stricter threshold of $P < 8 \times 10^{-10}$. These include 26 novel associations with measures of sleep quality and 10 with nocturnal sleep duration. The majority of identified variants associate with a single sleep trait, except for variants previously associated with restless legs syndrome. For sleep duration we identify a missense variant (p.Tyr727Cys) in *PDE11A* as the likely causal variant. As a group, sleep quality loci are enriched for serotonin processing genes. Although accelerometer-derived measures of sleep are imperfect and may be affected by restless legs syndrome, these findings provide new biological insights into sleep compared to previous efforts based on self-report sleep measures.

[#]A full list of authors and their affiliations appears at the end of the paper.

Sleep is an important human function, but many aspects of the mechanisms that regulate it remain poorly understood. Adequate sleep is important for health and wellbeing, and changes in sleep quality, quantity and timing are strongly associated with several human diseases and psychiatric disorders[1–5]. Identifying genetic variants influencing sleep traits will provide new insights into the molecular regulation of sleep in humans and help to establish the genetic contribution to causal links between sleep and associated chronic diseases, such as diabetes and obesity[6–10].

Genome-wide association studies (GWAS) are an important first step towards the discovery of new biological mechanisms of complex traits. Previous large-scale genetic studies of sleep traits have relied on self-reported measures. For example, using questionnaire data from 47,180 individuals, the CHARGE consortium identified the first common genetic variant, near *PAX8*, robustly associated with sleep duration[11]. Subsequent studies in up to 128,286 individuals using the interim data release of the UK Biobank identified two additional sleep duration loci[12,13] and a parallel analysis of the full UK Biobank release of 446,118 individuals identified a total of 78 associated loci[14]. Genetic associations have also been identified for other self-reported sleep traits, including chronotype[12,15,16], insomnia, and daytime sleepiness[13,17–21].

Although the reported associations revealed relevant pathways related to mechanisms underlying sleep regulation, in large-scale studies self-report measures are typically based on a limited number of questions that only approximate a limited number of sleep traits and may be subject to bias related to an individual's perception and recall of sleeping patterns[22–26]. Polysomnography (PSG) is regarded as the gold standard method of quantifying nocturnal sleep traits, but it is impractical to perform in large cohorts. Additionally, PSG is relatively burdensome, since it involves the use of complex equipment and experimental settings to represent individual's habitual sleep, making it less suitable for measuring sleep over multiple nights and capturing inter-daily variability. By contrast, research-grade activity monitors (accelerometers), also known as actigraphy devices, are more objective and may provide cost-effective estimates of sleep for large studies. To date, studies of limited sample size have been performed and focussed on daytime activity[27,28]. The UK Biobank study is a unique resource collecting vast amounts of clinical, biomarker, and questionnaire data on ~500,000 UK residents. Of these, 103,000 participants wore activity monitors continuously for up to 7 days. This provides an opportunity to derive accelerometer-based estimates of sleep quality, quantity and timing and to assess the genetics of sleep traits.

In this study, we identify genetic variants associated with accelerometer-derived measures of sleep and rest-activity patterns and use them to further understand the biology of sleep. We use accelerometer data from the UK Biobank to extract estimates of sleep characteristics using a heuristic method previously compared against independent PSG and sleep-diary datasets. These estimates have previously been demonstrated to be correlated with polysomnography and sleep diaries[29,30]. We analyse a total of eight accelerometer-based measures of sleep and activity timing. These include measures representative of sleep quality, including sleep efficiency (sleep duration divided by the time between the start and end of the first and last nocturnal inactivity period, respectively) and the number of nocturnal sleep episodes. In addition, we derive measures of timing (sleep midpoint, timing of the least-active 5 h (L5), and timing of the most-active 10 h (M10)), and duration (diurnal inactivity and nocturnal sleep duration and variability). We present a GWAS in 85,670 UK Biobank participants and validate our findings in three independent studies. Our analysis primarily focuses on traits that cannot be captured, or are unavailable, from self-report sleep measures, and are likely to be underpowered for GWAS in studies with PSG data due to limited sample sizes.

## Results

**Sleep quality and quantity are uncorrelated with timing**. Descriptive statistics and correlations between the eight accelerometer-derived phenotypes are shown in Table 1 and Supplementary Table 1. We observed little phenotypic correlation ($R$) between measures of sleep timing and measures of nocturnal sleep duration and quality ($-0.10 \leq R \leq 0.12$). These negligible or limited correlations between timing and duration are consistent with data from self-reported chronotype and sleep duration ($R = -0.01$). We also observed limited correlation between sleep duration and sleep quality as represented by the number of nocturnal sleep episodes ($R = 0.14$) but observed a stronger correlation between sleep duration and sleep efficiency ($R = 0.57$). The correlations between self-reported sleep duration and accelerometer-derived sleep duration was 0.19 and between self-reported chronotype (morningness) and L5 timing was $-0.29$.

**Accelerometer-derived sleep pattern estimates are heritable**. To estimate the proportion of variance attributable to genetic factors for a given trait, we used BOLT-REML to estimate SNP-based heritability ($h^2_{SNP}$) (Table 2). The $h^2_{SNP}$ estimates ranged from 2.8% (95% CI 2.0%, 3.6%) for variation in sleep duration (defined as the standard deviation of accelerometer-derived sleep duration across all nights), to 22.3% (95% CI 21.5%, 23.1%) for number of nocturnal sleep episodes. For sleep duration, we observed higher heritability using the accelerometer-derived measure ($h^2_{SNP} = 19.0\%$, 95% CI 18.2%, 19.8%) in comparison to self-report sleep duration ($h^2_{SNP} = 8.8\%$, 95% CI 8.6%, 9.0%). The heritability estimates for sleep and activity timings (maximum $h^2_{SNP} = 11.7\%$, 95% CI

---

**Table 1 Descriptive statistics of sleep and activity measures derived from accelerometer data**

| Measure | Mean | S.D. | Min | Max | N |
|---|---|---|---|---|---|
| L5 timing (hours from previous midnight) | 27.32 | 1.07 | 12.29 | 35.35 | 85,830 |
| M10 timing (hours from previous midnight) | 13.70 | 1.21 | 0.26 | 23.44 | 85,723 |
| Sleep midpoint (hours from previous midnight) | 26.99 | 0.91 | 16.25 | 31.98 | 85,502 |
| Sleep duration mean (hours) | 7.30 | 0.86 | 3.00 | 11.87 | 85,502 |
| Sleep duration (SD; (hours)) | 0.93 | 0.57 | 0.00 | 7.26 | 85,068 |
| Sleep efficiency (%) | 76.18 | 7.18 | 28.74 | 100.00 | 85,502 |
| Number of sleep episodes | 17.25 | 3.59 | 5.14 | 29.86 | 85,502 |
| Diurnal inactivity duration | 0.97 | 0.68 | 0 | 9.21 | 85,502 |

Units for the midpoint of the least-active 5 h (L5), midpoint of most-active 10 h (M10), sleep duration, sleep duration variation (SD), sleep midpoint and diurnal inactivity are in hours. Sleep efficiency is a ratio and number of sleep episodes is a count

10.9%, 12.5%) were lower than for self-report chronotype ($h^2_{SNP}$ = 13.7%, 95% CI 13.3%, 14.0%)[31].

**Low genetic correlation between sleep duration estimates**. To quantify the genetic overlap between accelerometer-derived and self-reported sleep traits, we performed genetic correlation

analyses using LD-score regression as implemented in LD-Hub[32]. We observed strong genetic correlations of L5, M10 and sleep midpoint timing with self-report chronotype ($r_G > 0.79$), and weaker genetic correlation between accelerometer-derived versus self-reported sleep duration ($r_G = 0.43$). This observation may be due to differences in the genetic contribution to variation in self-reported versus accelerometer-derived sleep duration or differences in the accuracy of self-reported phenotypes.

**Forty-seven genetic associations identified for sleep traits**. To identify genetic loci associated with accelerometer-derived sleep traits, we performed a genome-wide association analysis of 11,977,111 variants in up to 85,670 individuals for the eight accelerometer-derived sleep traits. We identified 47 genetic associations across seven of the phenotypes at the standard GWAS threshold ($P < 5 \times 10^{-8}$). Among these associations, 20 reached a more stringent threshold of $P < 8 \times 10^{-10}$. We estimate that this threshold reflects a better type 1 error rate to account for the approximate number of independent genetic variants analysed[31] and the 8 accelerometer-based traits (Table 3 and Supplementary

| Table 2 Heritability estimates of derived sleep variables from BOLT-REML | | |
|---|---|---|
| **Sleep variable** | $h^2$ | **95% CI** |
| Sleep duration | 0.190 | 0.182–0.198 |
| Sleep duration variability (SD) | 0.028 | 0.020–0.036 |
| Number of nocturnal sleep episodes | 0.223 | 0.215–0.231 |
| Sleep efficiency | 0.130 | 0.122–0.138 |
| L5 timing | 0.117 | 0.109–0.125 |
| M10 timing | 0.087 | 0.079–0.095 |
| Sleep midpoint timing | 0.101 | 0.093–0.109 |
| Diurnal inactivity | 0.148 | 0.134–0.161 |

**Table 3 Summary statistics for 47 genetic associations identified in the UK Biobank reaching $P < 5 \times 10^{-8}$**

| TRAIT | SNP | Chr | BP (hg19) | EA/OA | Freq | BETA | SE | P | Gene region |
|---|---|---|---|---|---|---|---|---|---|
| L5 timing | rs1144566 | 1 | 182,569,626 | C/T | 0.970 | 0.096 | 0.014 | 8E-12 | RGS16/RNASEL |
| L5 timing | rs113851554 | 2 | 66,750,564 | T/G | 0.057 | 0.133 | 0.011 | 2E-35 | MEIS1[a] |
| L5 timing | rs12991815 | 2 | 68,071,990 | C/G | 0.424 | 0.029 | 0.005 | 2E-09 | C1D[a] |
| L5 timing | rs9369062 | 6 | 38,437,303 | A/C | 0.708 | 0.039 | 0.005 | 9E-14 | BTBD9[a] |
| L5 timing | rs4882315 | 12 | 38,458,906 | T/C | 0.507 | 0.027 | 0.005 | 2E-08 | CPNE8/ALG10B |
| L5 timing | rs12927162 | 16 | 52,684,916 | G/A | 0.277 | 0.029 | 0.005 | 3E-08 | TOX3[a] |
| M10 timing | rs1973293 | 12 | 38,679,575 | C/T | 0.481 | 0.029 | 0.005 | 1E-09 | CPNE8/ALG10B |
| Sleep duration | rs2660302 | 1 | 98,520,219 | A/T | 0.811 | 0.041 | 0.006 | 9E-12 | DPYD |
| Sleep duration | rs113851554 | 2 | 66,750,564 | G/T | 0.943 | 0.110 | 0.011 | 2E-25 | MEIS1[a] |
| Sleep duration | rs62158170 | 2 | 114,082,175 | G/A | 0.217 | 0.054 | 0.006 | 3E-21 | PAX8 |
| Sleep duration | rs17400325 | 2 | 178,565,913 | T/C | 0.958 | 0.066 | 0.012 | 2E-08 | PDE11A |
| Sleep duration | rs72828540 | 6 | 19,102,286 | T/C | 0.752 | 0.041 | 0.005 | 1E-13 | LOC101928519 |
| Sleep duration | rs9369062 | 6 | 38,437,303 | C/A | 0.292 | 0.033 | 0.005 | 2E-10 | BTBD9[a] |
| Sleep duration | rs2975734 | 8 | 10,090,097 | C/G | 0.561 | 0.027 | 0.005 | 1E-08 | MSRA |
| Sleep duration | rs13282541 | 8 | 41,723,550 | C/T | 0.739 | 0.032 | 0.005 | 4E-09 | ANK1 |
| Sleep duration | rs2880370 | 8 | 105,987,057 | A/T | 0.670 | 0.028 | 0.005 | 2E-08 | LRP12/ZFPM2 |
| Sleep duration | rs800165 | 12 | 67,645,219 | C/T | 0.343 | 0.028 | 0.005 | 3E-08 | CAND1 |
| Sleep duration | rs10138240 | 14 | 63,353,479 | G/C | 0.514 | 0.029 | 0.005 | 7E-10 | KCNH5 |
| Sleep midpoint | rs11892220 | 2 | 231,691,067 | T/A | 0.339 | 0.029 | 0.005 | 3E-08 | CAB39 |
| Sleep efficiency | rs113851554 | 2 | 66,750,564 | G/T | 0.943 | 0.101 | 0.011 | 5E-22 | MEIS1[a] |
| Sleep efficiency | rs62158169 | 2 | 114,081,827 | T/C | 0.216 | 0.032 | 0.006 | 2E-08 | PAX8 |
| Sleep efficiency | rs17400325 | 2 | 178,565,913 | T/C | 0.958 | 0.074 | 0.012 | 2E-10 | PDE11A |
| Sleep efficiency | rs13094687 | 3 | 52,450,043 | G/A | 0.315 | 0.029 | 0.005 | 1E-08 | PHF7 |
| Sleep efficiency | rs13080973 | 3 | 138,596,050 | G/A | 0.202 | 0.032 | 0.006 | 3E-08 | FOXL2 |
| No. sleep episodes | rs12714404 | 2 | 282,462 | T/G | 0.283 | 0.037 | 0.005 | 1E-12 | ACP1/SH3YL1 |
| No. sleep episodes | rs310727 | 3 | 4,336,589 | T/C | 0.475 | 0.026 | 0.005 | 3E-08 | SUMF1/SETMAR |
| No. sleep episodes | rs55754932 | 3 | 87,847,754 | C/A | 0.284 | 0.037 | 0.005 | 2E-12 | HTR1F |
| No. sleep episodes | rs9864672 | 3 | 137,076,353 | T/C | 0.522 | 0.029 | 0.005 | 2E-10 | IL20RB/SOX14 |
| No. sleep episodes | rs4974697 | 4 | 2,473,092 | T/A | 0.390 | 0.026 | 0.005 | 5E-08 | RNF4 |
| No. sleep episodes | rs7377083 | 4 | 102,708,997 | A/C | 0.430 | 0.029 | 0.005 | 2E-09 | BANK1 |
| No. sleep episodes | rs749100 | 5 | 63,307,862 | A/G | 0.582 | 0.033 | 0.005 | 9E-12 | HTR1A/RNF180 |
| No. sleep episodes | rs9341399 | 6 | 73,773,644 | C/T | 0.936 | 0.066 | 0.010 | 6E-12 | KCNQ5 |
| No. sleep episodes | rs1889978 | 6 | 124,771,233 | C/T | 0.485 | 0.027 | 0.005 | 5E-09 | NKAIN2 |
| No. sleep episodes | rs2141277 | 7 | 39,099,178 | A/G | 0.478 | 0.026 | 0.005 | 1E-08 | POU6F2 |
| No. sleep episodes | rs10233848 | 7 | 103,122,645 | G/A | 0.293 | 0.035 | 0.005 | 2E-11 | RELN |
| No. sleep episodes | rs1124116 | 10 | 99,371,147 | A/G | 0.730 | 0.031 | 0.005 | 2E-09 | HOGA1/MORN4 |
| No. sleep episodes | rs4755731 | 11 | 43,685,168 | G/A | 0.431 | 0.028 | 0.005 | 3E-09 | HSD17B12 |
| No. sleep episodes | rs3751837 | 16 | 3,583,173 | C/T | 0.781 | 0.033 | 0.006 | 4E-09 | CLUAP1 |
| No. sleep episodes | rs8045740 | 16 | 20,262,776 | G/T | 0.868 | 0.052 | 0.007 | 6E-14 | GPR139 |
| No. sleep episodes | rs11078917 | 17 | 37,746,359 | A/C | 0.279 | 0.029 | 0.005 | 3E-08 | NEUROD2 |
| No. sleep episodes | rs11082030 | 18 | 35,501,739 | T/C | 0.725 | 0.030 | 0.005 | 8E-09 | CELF4 |
| No. sleep episodes | rs8098424 | 18 | 52,458,218 | G/A | 0.619 | 0.027 | 0.005 | 1E-08 | RAB27B |
| No. sleep episodes | rs76753486 | 19 | 42,684,264 | T/C | 0.084 | 0.047 | 0.008 | 2E-08 | DEDD2/ZNF526 |
| No. sleep episodes | rs429358 | 19 | 45,411,941 | T/C | 0.848 | 0.036 | 0.007 | 4E-08 | APOE |
| No. sleep episodes | rs12479469 | 20 | 61,145,196 | A/G | 0.342 | 0.031 | 0.005 | 4E-10 | MIR133A2 |
| Diurnal inactivity | rs17805200 | 9 | 13,764,434 | C/T | 0.272 | 0.031 | 0.005 | 5E-09 | MPDZ/NFIB |
| Diurnal inactivity | rs7155227 | 14 | 63,365,094 | T/G | 0.523 | 0.033 | 0.005 | 2E-12 | KCNH5 |

*CHR* chromosome, *BP* base-pair position (GRCh37/hg19), *EA/OA* effect allele/other allele, *Freq* effect allele frequency, *SE* standard error, *L5 timing* midpoint of least-active 5 h, *M10 timing* midpoint of most-active 10 h, *No. sleep episodes* number of nocturnal sleep episodes
[a]Locus previously reported for restless legs syndrome[35]

Figs. 1 and 2). Twenty-six associations were observed for sleep quality measures, including 21 variants associated with number of nocturnal sleep episodes and five associated with sleep efficiency (8 and 2 at $P < 8 \times 10^{-10}$, respectively). An additional eight genetic associations were identified for sleep and activity timing. These included six associated with L5 timing, one associated with M10 timing, and one associated with midpoint sleep. Only three associations with L5 timing were detected at $P < 8 \times 10^{-10}$. Finally, for sleep duration we observed 13 associations—11 for sleep duration and 2 associated with diurnal inactivity (6 and 1 at $P < 8 \times 10^{-10}$, respectively). Of these 47 associations reaching $P < 5 \times 10^{-8}$ and the 20 associations reaching $P < 8 \times 10^{-10}$, 31 and 9 were not previously reported in studies based on self-report measures, respectively (Table 3). The variance explained by all the discovered loci ranged from 0.04% for sleep midpoint timing to 0.8% for number of nocturnal sleep episodes. The lambda GC observed across these analyses ranged from 1.03 (sleep duration variability) to 1.14 (number of nocturnal sleep episodes), while LD-score intercepts ranged from 1.03 (diurnal inactivity) to 1.07 (sleep midpoint timing). Given the median $\chi^2$ test-statistic can be inflated for polygenic traits as sample size increases, the LD-score intercepts suggest limited inflation of test statistics observed is more likely to be due to the polygenicity of the phenotype tested over and above population stratification[33,34].

**Replication of 47 genetic associations in 5819 individuals**. We attempted to replicate our findings in up to 5819 adults from the Whitehall II ($N = 2,144$), CoLaus ($N = 2,257$), and Rotterdam Study (subsample from RS-I, RS-II and RS-III, $N = 1,418$) who had worn similar wrist-worn accelerometer devices for a comparable duration as the UK Biobank participants. Individual study and meta-analysis results for the three replication studies are presented in Supplementary Data 1. Of the 47 associations, the signal near *GPR139* (rs8045740) reached Bonferroni significance ($P = 0.001$) and 11 were associated at $P < 0.05$ after meta-analysis of the replication studies. Given the limited power to detect single SNP associations in the replication meta-analysis, we next examined the directional consistency of allele effect estimates. Of the 20 associations reaching $P < 8 \times 10^{-10}$, 18 were directionally consistent in the replication cohort meta-analyses ($P_{binomial} = 3 \times 10^{-4}$). Of the additional 27 signals, 18 were directionally consistent in the replication meta-analysis ($P_{binomial} = 0.03$). Finally, for traits with more than one independent lead SNP associated at $P < 5 \times 10^{-8}$ in the UK Biobank (Table 3), we combined the effects of the lead SNPs on the respective sleep trait (aligned to the trait increasing allele) and tested them in the replication data. In the combined-effects analysis, we observed overall associations with sleep duration ($P = 0.008$), sleep efficiency ($P = 3 \times 10^{-4}$), number of nocturnal sleep episodes ($P = 2 \times 10^{-6}$), and sleep timing ($P = 0.034$) (Supplementary Data 2).

**The genetics of sleep quality overlaps with sleep disorders**. Of the five variants associated with sleep efficiency, a measure of sleep quality, one was the strongly associated *PAX8* sleep duration signal[11] (rs62158169, $P = 2 \times 10^{-8}$) and one was a restless legs syndrome/insomnia-associated signal (*MEIS1*)[18,35] (rs113851554, $P = 5 \times 10^{-22}$). Of the 20 loci associated with number of nocturnal sleep episodes, one is represented by the *APOE* variant (rs429358). This variant is a proxy for the *APOE* ε4 risk allele that is strongly associated with late-onset Alzheimer's disease and cognitive decline[36]. The ε4 allele is associated with a reduced number of nocturnal sleep episodes (−0.13 sleep episodes; 95% CI: −0.16, −0.11; $P = 4 \times 10^{-8}$). This finding is strengthened by additional analyses of the ε2, ε3 and ε4 *APOE* Alzheimer's disease risk alleles, with an overall reduction in the number of nocturnal

sleep episodes observed with higher risk haplotypes (F(5, 72,578) = 5.36, $P = 0.001$) (Supplementary Table 2). This finding is inconsistent with the observational association between cognitive decline in older age and poorer sleep quality[37–40]. One possible explanation for this finding is ascertainment bias in the UK Biobank whereby carriers of ε4 risk allele are protected from cognitive decline through other factors. We also noted that the *APOE* ε4 risk allele was nominally associated ($P < 0.05$) with sleep timing (L5, −1.8 min per allele, $P = 4 \times 10^{-6}$), sleep midpoint (−0.6 min per allele; $P = 0.002$), sleep duration (−1.1 min per allele, $P = 7 \times 10^{-4}$), and diurnal inactivity (−1.0 min per allele, $P = 2 \times 10^{-5}$). Apart from the *APOE* variant (rs429358), which had double the effect size in the older half of the cohort (Supplementary Table 2), there were minimal differences in effect sizes in a range of sensitivity analyses, including removing individuals on sleep or depression medication, adjustments for BMI and lifestyle factors, and splitting the cohort by median age (Supplementary Data 3 and Supplementary Methods).

**Six associations identified for estimates of sleep timing**. We identified six loci associated with L5 timing, of which three have not previously been associated with self-report chronotype but have been associated with restless legs syndrome[35]. The lead variants at these three loci are in strong to modest LD with the previously reported variants associated with restless legs syndrome (rs113851554, *MEIS1*, $P = 2 \times 10^{-35}$, LD $r^2 = 1.00$; rs12991815, *C1D*, $P = 2 \times 10^{-9}$, LD $r^2 = 0.96$; rs9369062, *BTBD9*, $P = 9 \times 10^{-14}$, LD $r^2 = 0.49$). The three variants that reside in loci previously associated with self-report chronotype are in strong to modest linkage disequilibrium with those previously reported[12,15,16] (rs1144566, *RSG16*, $P = 8 \times 10^{-12}$, LD $r^2 > 0.91$; rs12927162, *TOX3*, $P = 3 \times 10^{-8}$, LD $r^2 = 1.00$; rs4882315, *ALG10B*, $P = 2 \times 10^{-8}$, LD $r^2 = 0.58$). The variant rs1144566 is a missense coding change (p.His137Arg) in exon 5 of *RSG16*, a known circadian rhythm gene, which contains variants strongly associated with self-report chronotype[12]. In a parallel self-report chronotype study in the UK Biobank, rs1144566 represented the strongest association, with the T allele having a morningness odds ratio of 1.26 ($P = 2 \times 10^{-95}$)[31]. In addition, variants in the region of *TOX3* have previously been associated with restless legs syndrome[35]. However, our lead SNP (rs12927162) was not in LD with the previously reported index variant at this locus (rs45544231, LD $r^2 = 0.004$). There were minimal differences in effect sizes when we performed a range of sensitivity analyses, including removing individuals on depression medication, adjustments for BMI and lifestyle factors and splitting the cohort by median age (Supplementary Data 3 and Supplementary Methods).

**Ten novel loci associated with estimates of sleep duration**. We identified 11 loci associated with accelerometer-derived sleep duration, including ten not previously reported to be associated with self-report sleep duration, despite the fivefold increase in sample size available for a parallel self-report sleep duration GWAS study[14] (Fig. 1 and Supplementary Data 4). This lower overlap in signals is consistent with the lower genetic correlation between self-reported and accelerometer-derived sleep duration than between chronotype and accelerometer-derived measures of sleep and activity timing. The lead variants representing the ten new sleep duration loci all had the same direction and larger effects in the accelerometer data compared to self-report data, with effect sizes ranging from 1.3 to 5.9 min compared to 0.1 to 0.8 min (self-report $P < 0.05$), with the *MEIS1* locus having the strongest effect. Two of the ten new sleep duration signals, rs113851554 in *MEIS1* ($P = 2 \times 10^{-25}$) and rs9369062 in *BTBD9*

$(P = 2 \times 10^{-10})$, have previously been associated with restless legs syndrome. The one variant previously detected based on self-report sleep duration, near *PAX8*, was the first variant to be associated with sleep duration through GWAS[11]. The minor *PAX8* allele effect size was consistent across accelerometer-derived measures of sleep duration (2.7 min per allele, 95% CI: 2.1 to 3.3, $P = 3 \times 10^{-21}$) and self-report sleep duration (2.4 min per allele, 95% CI: 2.1 to 2.8, $P = 7 \times 10^{-49}$). We observed similar effect sizes in a subset of 72,510 unrelated Europeans from the UK Biobank, when removing individuals on depression medication and after adjusting for BMI and lifestyle factors. To confirm that associations were not influenced by age-related differences in sleep, we confirmed that there was also no difference in effect sizes between younger and older individuals (above and below the median age of 63.7 years) (Supplementary Data 3).

**Fine-mapping analysis identifies likely causal variants**. To identify credible SNP sets likely to contain causal variants within 500 Kb of lead SNPs for each trait with a genetic association ($P < 5 \times 10^{-8}$) we used FINEMAP[41] to identify credible sets of likely causal SNPs ($\log_{10}$ Bayes Factor > 2) (Supplementary Data 5). This

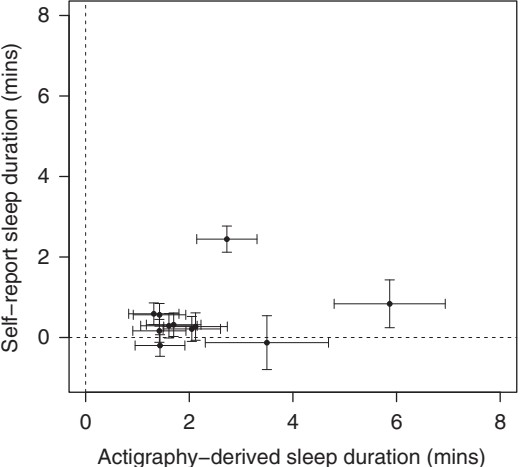

**Fig. 1** Comparison of SNP effect estimates on accelerometer and self-report sleep duration. The effects for 11 genetic variants associated with accelerometer-derived sleep duration against effect estimates from a parallel GWAS of self-report sleep duration[14] are presented. Error bars represent the 95% confidence intervals for each effect estimate

approach places a probability on the likelihood that a variant, among those tested, represents the causal allele. Two loci contained a coding variant with a probability >80% for being the causal variant. The first variant (rs17400325, MAF = 4.2%) was a missense variant (p.Tyr727Cys) in *PDE11A*, a phosphodiesterase highly expressed in the hippocampus that was associated with sleep duration ($P = 2 \times 10^{-8}$) and sleep efficiency ($P = 2 \times 10^{-10}$). The other was the missense *APOE* variant, a proxy for the ε4 allele known to predispose to Alzheimer's disease and responsible for the association signal with the number of nocturnal sleep episodes. Of the remaining loci, five fine-mapped variants are eQTLs in the Genotype-Tissue Expression (GTEx) project[42]. Of these only the fine-mapped variant at the *CLUAP1* locus associated with the number of nocturnal sleep episodes ($P = 4 \times 10^{-9}$) was the lead variant for the corresponding eQTL ($\log_{10}$ Bayes Factor = 2.48, $P_{causal} = 0.72$) (Supplementary Data 5). *CLUAP1* has been previously associated with photoreceptor maintenance[43].

**Serotonin pathway-related genes enriched at associated loci**. We used MAGMA[44] to assess tissue enrichment of genes at associated loci across the sleep traits. All traits showed an enrichment of genes in the cerebellum (Supplementary Figs. 3 and 4). Loci associated with number of nocturnal sleep episodes were enriched for genes involved in serotonin pathways ($P_{Bonferroni} = 3 \times 10^{-4}$) (Supplementary Table 3).

**Associated variants are implicated in restless legs syndrome**. We observed most variants to be associated with either sleep quality, duration, or timing, but not combinations of these sleep characteristics. However, the variant rs113851554 at the *MEIS1* locus was associated with sleep quality (sleep efficiency), duration, and timing (L5). In addition, the variant rs9369062 at the *BTBD9* locus was associated with both sleep duration and L5 timing. Both variants have previously been reported as associated with restless legs syndrome (Fig. 2). To follow up this observation, we performed Mendelian Randomisation using 20 variants associated with restless legs syndrome in the discovery stage of the most recent and largest genome-wide association study[35]. We tested these 20 variants against all eight activity-monitor-derived sleep traits and showed a clear causative association of restless legs syndrome with all sleep traits. We also observed a causative association of restless legs syndrome with self-report sleep duration and chronotype, suggesting that variants associated with restless legs syndrome were not artefacts of the accelerometer-derived measures of sleep (Supplementary Data 6).

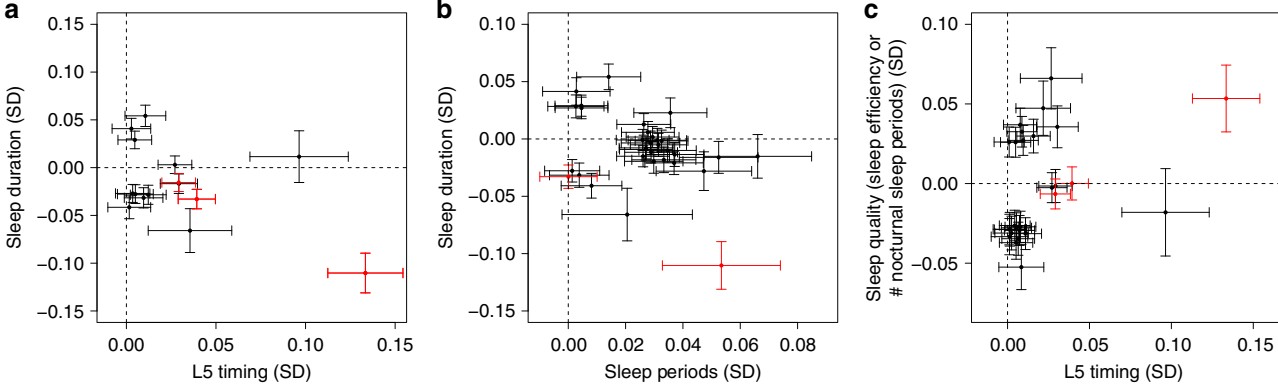

**Fig. 2** Effects of restless legs syndrome-associated SNPs on derived sleep traits. Presented are the effect estimates for genetic variants associated with **a** either L5 timing or sleep duration, **b** either sleep duration or the number of nocturnal sleep episodes, and **c** either L5 timing or sleep quality (number of nocturnal sleep episodes or sleep efficiency). Variants previously associated with restless legs syndrome are highlighted in red. Effect estimates represent standard deviations of the inverse-normal distribution of each trait. Error bars represent the 95% confidence intervals for each effect estimate

**Waist-hip-ratio causally influences sleep outcomes**. To assess causality of phenotypes, we used genetic correlations to prioritise traits with evidence of genetic overlap for subsequent Mendelian Randomisation analyses using LD-Hub[32]. We tested for genetic correlations between the eight activity-monitor-derived measures and 234 published GWAS studies across a range of diseases and traits. Given previous reports that genetic correlations are similar to phenotypic correlations[45], this approach also enabled us to analyse phenotypes under-represented, not recorded, or not well defined within the UK Biobank. After adjustment for the number of genetic correlations tested ($8 \times 234$), we observed genetic correlations between sleep traits and obesity and educational attainment related traits (Supplementary Data 7). After adjusting for the number of tests in the bi-directional MR analysis (100), we observed evidence that higher waist-hip-ratio (adjusted for BMI) is causally associated with lower sleep duration ($P_{\mathrm{IVW}} = 5 \times 10^{-6}$) and lower sleep efficiency ($P_{\mathrm{IVW}} = 3 \times 10^{-4}$). In addition, we

observed higher educational attainment to be causally associated with lower sleep duration ($P_{\mathrm{IVW}} = 5 \times 10^{-5}$). However, given the genetic correlation and MR analyses are not independent, only the causal association of waist-hip-ratio (adjusted for BMI) on sleep duration remained significant after applying a more stringent threshold ($P_{\mathrm{IVW}} \leq 3 \times 10^{-5}$) to account for a maximum of 234 bi-directional MR analyses (Supplementary Data 8). We observed no evidence of causal effects of accelerometer-based sleep traits on outcomes tested (Supplementary Data 9).

**Self-report and accelerometer sleep traits effects correlate**. We compared effects of variants associated with self-reported sleep duration and chronotype identified in parallel GWAS analyses. Overall, we observed directional consistency with the accelerometer-derived measures. In a parallel GWAS of self-reported sleep duration in 446,118 individuals from the UK Biobank[14], we identified 78 associated loci at $P < 5 \times 10^{-8}$. Sixty-seven (85.9%) of these SNPs were directionally consistent between the self-report and activity-monitor-derived sleep duration GWAS ($P_{\mathrm{binomial}} = 6 \times 10^{-11}$; Fig. 3). Furthermore, in a parallel report[31] we have shown that of the 341 lead variants at self-reported chronotype loci, 310 (90.9%) had a consistent direction of effect for accelerometer-derived mid-point-sleep ($P_{\mathrm{binomial}} = 5 \times 10^{-59}$), 316 (92.7%) with L5 timing ($P_{\mathrm{binomial}} = 3 \times 10^{-65}$) and 310 (90.9%) with M10 timing ($P_{\mathrm{binomial}} = 5 \times 10^{-59}$). Figure 4 shows a scatter plot of self-reported associated chronotype effects against L5 timing effects.

## Discussion

Our analysis presents the largest-scale GWAS of multiple sleep traits estimated from accelerometer data using our activity-monitor sleep algorithm, with estimates previously demonstrated to be highly correlated with polysomnography[29,30]. We have identified 47 genetic associations at $P < 5 \times 10^{-8}$ across seven traits representing sleep duration, quality and timing. These loci included ten novel variants for sleep duration and 26 for sleep quality not detected in larger studies of self-reported sleep traits.

Of the associated loci, a low-frequency (MAF = 4.2%) missense variant (p.Tyr727Cys) at the *PDE11A* locus (rs17400325) was associated with sleep duration and sleep efficiency. The variant was associated with sleep duration ($P = 0.004$) in the meta-analysis of the replication cohort. Fine-mapping provided a high

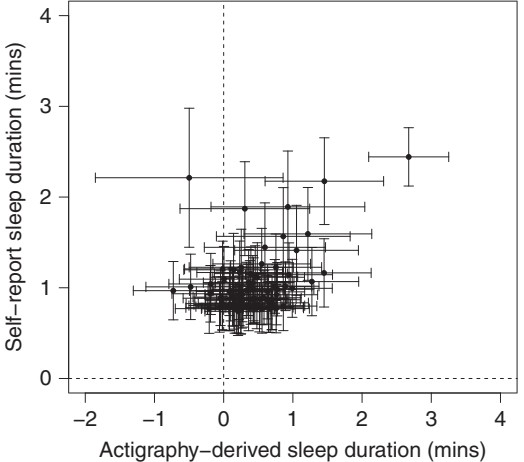

**Fig. 3** Comparison of effect estimates for SNPs associated with self-reported sleep duration. The effect estimates for 78 genetic variants associated with self-report sleep duration in a parallel GWAS effort[14] are plotted against the effect estimates on accelerometer-derived estimates of sleep duration. Error bars represent the 95% confidence intervals for each effect estimate

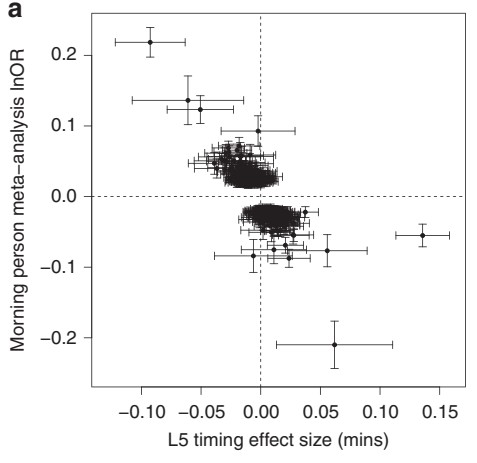
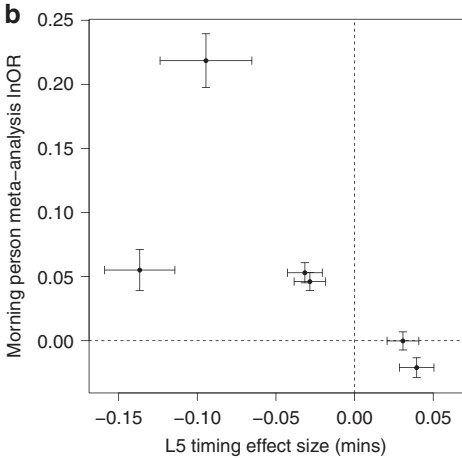

**Fig. 4** Comparisons of genetic effect estimates on morning person (binary chronotype) and L5 timing. Plotted are the genetic effect estimates for variants associated with chronotype based on the latest self-report chronotype meta-analysis[31] against accelerometer-derived estimates of L5 timing: **a** Three-hundred fifty-one variants identified from the self-report chronotype meta-analysis and **b** six variants identified for L5 timing. Error bars represent the 95% confidence intervals for each effect estimate. Chronotype effect estimates are reported for variants identified in the primary sample size meta-analysis using effect sizes (lnOR morningness) derived in the secondary effect size meta-analysis

probability (>90%) that this is the causal variant at the locus. This variant has previously been associated with migraine and near-sightedness (myopia) in a scan of 42 traits from 23andMe[46]. In the UK Biobank the variant was not associated with migraine ($P = 0.44$), consistent with the latest migraine meta-analysis where it was not among the associated loci[47], but was associated with myopia ($P = 9 \times 10^{-10}$). The allele that associates with reduced risk of myopia is associated with increased sleep efficiency and duration. Protein truncating variants in *PDE11A* have been suggested to cause adrenal hyperplasia;[48] however, one of these variants (R307X, rs76308115) is present at 0.5% frequency in the UK Biobank (with 11 rare allele homozygotes) and is not associated with sleep efficiency ($P = 0.99$) or duration ($P = 0.54$). This suggests that if Tyr727Cys *PDE11A* is the causal variant at this locus then it is an activating mutation. *PDE11A* is expressed in the hippocampus and it has been suggested as a potential biological target for interventions in neuropsychiatric disorders[49].

Our analysis identified variants in loci that were enriched for genes involved in the serotonin pathway—the strongest pathway associated with sleep quality. Serotonergic transmission plays an important role in sleep cycles[50,51]. High levels of serotonin are associated with wakefulness and lower levels with sleep. Furthermore, serotonin is synthesised by the pineal gland as a processing step for melatonin production, a key hormone in circadian rhythm regulation and sleep timing[52].

A subset of variants previously associated with restless legs syndrome were associated with sleep duration, quality and timing measures. In the UK Biobank, restless legs syndrome was only identified through the Hospital Episodes Statistics (HES) data using the ICD-10 code G25.8 (Other specified extrapyramidal and movement disorders), the parent category of the more specific G25.81 code (Restless legs syndrome). Under the assumption that all individuals reporting G25.8 had restless legs, we observed 38 individuals within our accelerometer subset. Removing these individuals did not change our conclusions. Given that the same variants are also associated with self-report measures of sleep duration, chronotype and insomnia, this observation may not be an artefact caused by limb movements during sleep. On the other hand, the repetitive periodic limb movements (PLMS) that people with RLS typically experience during sleep could have been detected by the accelerometers and confounded the parameters. Studies with more in-depth phenotyping of sleep disorders are needed to more fully evaluate the contribution of RLS and PLMS to sleep traits, especially in light of a recent paper showing that associations with *MEIS1* were only in those with RLS[53].

Our Mendelian Randomisation analysis also provides some evidence of a causative effect of higher waist-hip-ratio (adjusted for BMI) on lower sleep duration and lower sleep efficiency. This suggests that fat distribution plays a role in sleep, although there was also a nominal causative association with BMI, which also suggests a general role of overall adiposity. We also observed evidence of a causative association between higher educational attainment and lower sleep duration. Both the adiposity and educational attainment MR results were robust to a range of MR sensitivity analyses (Supplementary Data 8). We did not observe evidence of a causal effect of accelerometer-derived sleep variables on genetically correlated traits. This may be due to the relatively limited power because of the relatively small number of genetic instruments available.

Our data provide strong evidence that some accelerometer-derived measures of sleep provide higher precision than self-report measures, while for others there is little gain through accelerometer-derived measurement with questionnaire data being just as effective. For example, of the 11 accelerometer-based sleep duration loci we identified, only one (the *PAX8* variant) had been previously identified in self-reported sleep duration GWAS

despite these studies having much larger sample sizes. Variants with nominal evidence of association with self-reported sleep duration had weaker effects. This difference may be due to reporting biases related to the UK Biobank questionnaire (e.g., response was in hourly increments) and due to asking participants to include nap-time in their sleep duration. In contrast the accelerometer-derived estimates of L5 timing, the least-active 5 h of the day, correlated well with self-report estimates. These data suggest that the answer to the very simple question "are you a morning or evening person" provides similar power as wearing accelerometers for 7 days and nights. In a parallel GWAS analysis, the *PAX8* variant was also associated with self-report insomnia[20]. In addition, five of the loci were nominally associated ($P < 0.05$) with either self-report sleep duration or insomnia. At least two of the sleep duration signals have been previously associated with mental health disorders, including schizophrenia and migraine[46,54].

The Alzheimer's disease risk allele at the *APOE* locus had apparently paradoxical associations with sleep related traits. Given the well-established association between the *ε4* allele and greater risk of Alzheimer's disease, we would not expect associations between this allele and higher sleep quality given previous associations of adverse sleeping patterns with cognitive decline and Alzheimer's disease[4]. A similar paradoxical association was also reported recently in a study of over 2,300 men aged over 65 with overnight PSG data that showed the total time in stage N3 sleep was higher for individuals carrying two copies of *ε4* compared with those carrying one or zero copies[55]. Furthermore, a recent genetic study of physical activity also identified a paradoxical association between the *ε4* allele and increased levels of physical activity[56]. The more likely explanations for these associations we suggest are ascertainment and survival bias. The UK Biobank participants ranged from 44 to 79 years of age when wearing the accelerometer devices. Older UK Biobank participants, with the highest risk of cognitive decline with an *ε4/ε4* haplotype and agreeing to an accelerometer-based experiment could be protected from cognitive decline because of selection bias due to other factors[57]. To participate in the UK Biobank study and agree to accelerometer data collection several years after study baseline is less likely to occur in individuals who are in cognitive decline. As a result, the *ε4* risk allele may present an association with higher sleep quality. Consistent with this potential bias, the *ε4* allele association with reduced numbers of nocturnal sleep episodes is stronger in older age. For example, when splitting individuals by median age, the per allele effect on number of sleep episodes was twice that of the older versus younger group.

There are some limitations to this study. First, a sleep diary was not collected by the UK Biobank participants, a traditional tool to guide the start and end timing of nocturnal sleep episodes, commonly used in actigraphy studies. We have developed and used an open source method to overcome the lack of a sleep diary that has been compared against polysomnography[29,30] to estimate sleep onset and waking up time. However, as no sleep diary data exists it is hard to define bedtime prior to sleep, resulting in the inability to characterise phenotypes such as sleep onset latency (the time between going to bed and falling asleep). Second, the activity monitors were worn up to 10 years from when baseline data was collected. Despite this, the correlation between self-report and activity measures of sleep duration was consistent with previous studies, and the correlation did not differ based on time between baseline (self-report time) and accelerometer wear when splitting by time-difference deciles ($r = -0.03$, $P = 0.94$). Third, due to relatively small sample sizes of replication studies, we had limited power to replicate associations identified in the UK Biobank. The variance explained by individual variants in the

UK Biobank ranged from 0.03% to 0.19%, for which we had <63% power to detect at a statistical threshold of $P = 0.001$ (accounting for 47 tests) in the meta-analysis of 4,401 individuals. However, we observed an enrichment for directional consistency in effect estimates in the replication meta-analysis and in combined-effects analyses identified associations for sleep duration, sleep efficiency, number of nocturnal sleep episodes and sleep timing. Fourth, the UK Biobank participants are not representative of the UK population, as participants had a higher socio-economic position overall and were healthier, on average, given the prevalence of diseases among the participants[57,58]. This was particularly true of the participants who took part in the activity-monitor study. Finally, it is important to keep in mind that while accelerometry provides a more objective means of assessing sleep and wake than self-report, it has its own limitations. Measures of sleep using actigraphy are intrinsically difficult to interpret, as awake and not moving cannot be distinguished from sleep. Furthermore, although small studies have shown limited effects of events that disturb sleep (such as respiratory events or periodic limb movements), the effect on large-scale data is difficult to assess. Moderate to severe sleep apnea (≥15 apneas or hypopneas per hour of sleep) and periodic limb movements in sleep (≥15 movements per hour of sleep) are relatively common in individuals within the age range of the present study[59,60]. Unfortunately, these conditions were not captured well in the UK Biobank study, limiting the possibility of evaluating the effects of such sleep disorders on accelerometer-derived sleep traits. Future studies of PSG-derived metrics of sleep, such as total sleep duration, sleep efficiency, and proportions of sleep stages should be conducted. For people with insomnia, accelerometry tends to overestimate sleep because time spent lying still in bed awake attempting to sleep can be scored as sleep[61]. However, most studies have relied on a devices that measure a single axis of movement that could be more prone to these errors, and our recent work suggests that newer triaxial devices may be more accurate[30].

In conclusion, we have performed the most comprehensive GWAS of accelerometer-derived sleep measures to date. We demonstrated that self-report measures are good proxies for accelerometer-derived sleep measures. However, through the investigation of accelerometer-derived sleep measures we found additional loci not identified by previous self-report GWAS studies. These loci harbour likely causal variants associated with poor sleep.

## Methods

**UK Biobank participants.** The study population was drawn from the UK Biobank study—a longitudinal population-based study of individuals living in the UK[58]. Analyses were based on individuals estimated to be of European ancestry. European ancestry was defined through the projection of UK Biobank individuals into the principal component space of the 1000 Genomes Project samples[62] and subsequent clustering based on a $K$-means approach, centring on the means of the first four principal components.

**Genetic data.** Imputed genetic data was downloaded from the UK Biobank[63]. We limited our analysis to 11,977,111 genetic variants imputed using the Haplotype Reference Consortium imputation reference panel with a minimum minor allele frequency (MAF) > 0.1% and imputation quality score (INFO) > 0.3.

**Activity-monitor devices.** A triaxial accelerometer device (Axivity AX3) was worn between 2.8 and 9.7 years after study baseline by 103,711 individuals from the UK Biobank for a continuous period of up to 7 days. Data collection and initial quality checks were performed centrally by members of the UK Biobank study[64]. Of these 103,711 individuals, we excluded 11,067 individuals based on activity-monitor data quality. This included individuals flagged by UK Biobank as having data problems (field 90002), poor wear time (field 90015), poor calibration (field 90016), or unable to calibrate activity data on the device worn itself requiring the use of other data (field 90017). Individuals were also excluded if number of data recording errors (field 90182), interrupted recording periods (field 90180), or duration of interrupted recoding periods (field 90181) was greater than the respective variable's 3rd

quartile + 1.5 × IQR. Phenotypes determined using the SPT-window (all phenotypes except L5 and M10 timing) had additional exclusions based on short (<3 h) and long (>12 h) mean sleep duration and too low (≤5) or too high (≥30) mean number of sleep episodes per night (see below). These additional exclusions were to ensure that individuals with extreme (outlying), and likely incorrect, sleep characteristics were not included in any subsequent analyses. A maximum of 85,723 individuals remained for our analyses.

**Accelerometer data processing and sleep measure derivations.** We derived eight measures of sleep quality, quantity and timing. All measures were derived by processing raw accelerometer data (.cwa). We first converted the .cwa files available from the UK Biobank to .wav files using omconvert for signal calibration to gravitational acceleration[64,65] and interpolation[64]. The .wav files were processed with the open source R package GGIR[30] (https://doi.org/10.5281/zenodo.1175883 (Version v1.5-17)) to infer accelerometer non-wear time[66], and extract the $z$-angle across 5-s epochs from the time-series data for subsequent use in estimating the sleep period time window[30] and sleep episodes within it[29].

The sleep period time window (SPT-window) was estimated using an algorithm previously compared against PSG data and described[30] and implemented in the GGIR R package. Briefly, for each individual, median values of the absolute change in estimated $z$-angle (representing the dorsal-ventral direction when the wrist is in the anatomical position) across 5-min rolling windows were calculated across a 24-h period, chosen to make the algorithm insensitive to accelerometer orientation. The 10th percentile was incorporated into the threshold distinguishing movement from non-movement. Bouts of inactivity lasting ≥ 30 min are recorded as inactivity bouts. Inactivity bouts that are < 60 min apart are combined to form inactivity blocks. The start and end of the longest block defined the start and end of the SPT-window.

Sleep duration and variability were estimated based on sleep episodes within the STP-window. Sleep episodes within the SPT-window were defined as periods of at least 5 min with no change larger than 5° associated with the $z$-axis of the activity-monitor, as motivated and described in van Hees et al.[29]. The summed duration of all sleep episodes was used as indicator of sleep duration within the SPT-window. The total duration over the activity-monitor wear time was averaged. Individuals with an average sleep duration <3 h or >12 h were excluded from all analyses. In addition, the standard deviation of sleep duration was also calculated and put forward for statistical analysis for individuals with the maximum days ($N = 7$) of accelerometer wear.

Sleep efficiency was calculated as sleep duration (defined above) divided by the time elapsed between the start of the first inactivity bout and the end of the last inactivity bout (which equals the SPT-window duration).

The number of nocturnal sleep episodes was defined as the number of sleep episodes within the SPT-window. Individuals with an average number of nocturnal sleep episodes ≤5 or ≥30 were excluded from all analyses.

Least-active 5 h (L5) timing was defined as the midpoint of the least-active 5 h (L5) of each day. The least-active 5 h was defined as the 5-h period with the minimum average acceleration. These periods were estimated using a rolling 5-h time window. The midpoint was defined as the number of hours elapsed since the previous midnight (for example, 7 p.m. = 19 and 2 a.m. = 26). Days with <16 h of valid-wear time (as estimated by GGIR) were excluded from L5 estimates.

Most-active 10 h (M10) timing was defined as the midpoint of the most-active 10 h (M10) of each day. The most-active 10 h was defined as the 10-h period with the maximum average acceleration. These periods were estimated using a rolling 10-h time window. The midpoint was defined as the number of hours elapsed since the previous midnight. Days with < 16 h of valid-wear time (as estimated by GGIR) were excluded from M10 estimates.

Sleep midpoint timing was calculated for each sleep period as the midpoint between the start of the first detected sleep episode and the end of the last sleep episode used to define the overall SPT-window (above). This variable is represented as the number of hours from the previous midnight.

Diurnal inactivity was estimated by the total daily duration of estimated bouts of inactivity that fell outside of the SPT-window. This measure captures very inactive states such as napping and wakeful rest but not inactivity such as sitting and reading or watching television, which are associated with a low but detectable level of movement.

**Comparison against self-reported sleep measures.** We performed analyses of self-reported measures of sleep. Self-reported measures analysed included (a) the number of hours spent sleeping over a 24-h period (including naps); (b) insomnia; (c) chronotype—where "definitely a 'morning' person", "more a 'morning' than 'evening' person", "more an 'evening' than a 'morning' person", "definitely an 'evening' person" and "do not know", were coded as 2, 1, −1, -2 and 0, respectively, in our continuous variable.

**SNP-based heritability analysis.** We estimated the heritability of the eight derived accelerometer traits using BOLT-REML (version 2.3.1)[67]. We used 524,307 high-quality genotyped single nucleotide polymorphisms (SNPs) (bi-allelic; MAF ≥ 1%; HWE $P > 1 \times 10^{-6}$; non-missing in all genotype batches, total missingness <1.5% and not in a region of long-range LD[68]) to build the relatedness

model and thus to estimate heritability. For LD structure information, we used the default 1000 Genomes 'LD-score' table provided with the BOLT-REML software.

**Genome-wide association analyses.** We performed all association tests in the UK Biobank using BOLT-LMM v2.3[67], which applies a linear mixed model (LMM) to adjust for the effects of population structure and individual relatedness, and enables the inclusion of all related individuals in our white European subset, boosting our power to detect associations. This meant a sample size of up to 85,670 individuals, as opposed to a maximal set of 72,696 unrelated individuals. At runtime, all phenotypes were first adjusted for age at accelerometry (estimated using month of birth and date of first recording day), sex, study centre (categorical), season when activity-monitor worn (categorical) and genotyping array (categorical; UK Bileve array, UKB Axiom array interim release and UKB Axiom array full release). All phenotypes except sleep duration variation were also adjusted for the number of measurements used to calculate each participant's measure (number of L5/M10 measures for L5/M10 timing, number of days for diurnal inactivity and number of nights for all other phenotypes). The number of sleep episodes phenotype was further adjusted for time in bed (length of SPT-window). Phenotypes were analysed on their original-scale and the inverse-normal-scale after transformation and all results (except those that refer to interpretable effect sizes) are reported for the inverse-normal scale analyses.

**Replication of findings.** Associations reaching $P < 5 \times 10^{-8}$ were followed up in the CoLaus, Whitehall II and Rotterdam studies. The GENEActiv accelerometer was used by the CoLaus and Whitehall II studies and worn on the wrist by the participants. In the CoLaus study, 2967 individuals wore the accelerometer for up to 14 days. Of these, 10 were excluded because of insufficient data, 234 excluded as non-European, and a further 148 were excluded due to an average sleep duration of <3 h or >12 h. A total of 2575 individuals remained for analysis of which 2257 had genetic data. In the Whitehall II study, 2144 were available for analysis, with the GENEActiv accelerometer worn for up to 7 days having performed the same exclusions. The Rotterdam Study used the Actiwatch AW4 accelerometer device (Cambridge Technology Ltd.). All 24-h periods with more than three continuous hours missing were excluded from the analyses to prevent a time-of-day effect. Recordings were also excluded if consisting of <96 h ($n = 109$ excluded) or if collected in a week of daylight-saving time ($n = 26$), resulting in 1734 persons of which genetic data was available for 1418[69]. For all replication studies, the derivation of the sleep characteristics and the same overall- and trait-specific exclusion criteria outlined above applied. Where available, accelerometer-derived phenotypes were analysed both on the original-scale and inverse-normal scale. Genetic association analysis was based on imputed data (where available) and performed using standard multiple linear regression. The covariates incorporated into the model were the same as those used in the UK Biobank analysis. Overall summary statistics were obtained through inverse-variance-based meta-analysis implemented in METAL[70]. Combined variant effects on respective traits were subsequently calculated using the 'metan' function in STATA using the betas and standard errors obtained through the primary meta-analysis of the three replication studies.

**Gene-set, tissue enrichment and GWAS catalogue analyses.** Gene-set analyses and tissue expression analyses were performed using MAGMA[44] as implemented in the online Functional Mapping and Annotation of Genome-Wide Association Studies (FUMA) tool[71]. For lead and candidate SNP identification, the default settings were used: lead variants were required to have a minimum $P$-value of $5 \times 10^{-8}$; $r^2$ threshold for defining LD structure of lead variants was set to 0.6; the maximum $P$-value cutoff was set to 0.05; the reference panel population was chosen to be 1000 Genomes Phase 3; variants in the reference panel but not in the GWAS were included; the minimum minor allele frequency was set to 0.01 and a maximum allowed distance of 250 k between LD blocks was used for variants to be included in the same locus. For assigning variants to genes for gene-set enrichment analyses, positional mapping was used with variants assigned to a gene if they were within the gene start and end points (by setting the distance either side to 0 kb) and only protein-coding genes were included in the mapping process. MAGMA gene-set enrichment analysis, implemented in FUMA, adopts a competitive test of gene-set enrichment using 10,894 gene sets obtained from MSigDB[72]. Tested gene sets include BioCarta, REACTOME, KEGG and GO among others; a full list of gene sets used by MSigDB can be found at http://software.broadinstitute.org/gsea/msigdb/collections.jsp. Bonferroni correction for gene-set enrichment adjusted for the number of gene sets tested and was applied to each phenotype separately. Analysis of differentially expressed genes was based on data from GTEx v6 RNA-seq data[73]. Enrichment analyses of the overlap with associations previously reported through GWAS, using data from the GWAS Catalogue, was also implemented through FUMA. Enrichment $P$-values for the proportion of overlapping genes present was based on the NIH GWAS catalogue[74].

**Fine-mapping association signals.** Fine-mapping analyses were performed using FINEMAP v1.2[41] using the software's shotgun stochastic search function and by setting the maximum number of causal SNPs at each locus to 20. At each locus, we included only those with $P < 0.01$ and within 500 Kb either side of the lead variant to limit the number of SNPs in the analysis. We constructed the LD matrix by

calculating the Pearson correlation coefficient for all SNP–SNP pairs using SNP dosages. Dosages were derived from the unrelated European subset of the full UK Biobank imputed genotype probabilities ($N = 379,769$). We considered a SNP to be causal if it's $\log_{10}$ Bayes factor was >2, as recommended in the FINEMAP manual (http://www.christianbenner.com/index_v1.2.html).

**Alamut annotation and eQTL mapping.** We performed variant annotation of our fine-mapped loci using Alamut Batch v1.8 (Interactive Biosoftware, Rouen, France) using all default options and genome assembly GRCh37. For each annotated variant, we retained only the canonical (longest) transcript and reported the variant location, coding effect and the predicted local splice site effect. To investigate whether the fine-mapped SNPs were eQTLs, we searched for our SNPs in the single-tissue *cis*-eQTL and multi-tissue eQTL datasets (v7), available at the GTEx portal (https://www.gtexportal.org/home/datasets) for significant SNP-gene eQTL associations. In the multi-tissue eQTL data, we reported a SNP as an eQTL for a gene if the SNP-gene association was significant in the meta-analysis across all tissues. Using the single-tissue *cis*-eQTL data, we performed a lookup of our fine-mapped SNPs for significant SNP-gene associations in brain tissues only. For each gene with a fine-mapped SNP significantly associated with expression levels, we reported the tissue with the strongest evidence (lowest $P$-value) of association and the correlation ($r^2$) between the fine-mapped SNP and the tissue's strongest eQTL SNP.

**Sensitivity analyses.** To assess whether stratification was responsible for any of the individual variant associations in a subset of the cohort, we performed multiple sensitivity analyses in unrelated European subsets of the UK Biobank using STATA. Relatedness was inferred using kinship coefficients provided by the UK Biobank. The maximal set of unrelated individuals (<3rd degree) was put forward for sensitivity analyses. The sensitivity analyses carried out were: (1) males only, (2) females only (3) individuals younger than the median age (at start of the activity-monitor wear period), (4) individuals older than the median age, (5) adjustment for body mass index (BMI) (UK Biobank data field 21001), (6) adjusting for BMI and lifestyle factors and (7) excluding individuals working shifts, taking medication for sleep or psychiatric disorders, self-reporting a mental health or sleep disorder, or diagnosed with depression, schizophrenia, bipolar disorder, anxiety disorders or mood disorder in the HES data (see Supplementary Methods). Sensitivity analyses were performed using fixed-effect models. Phenotypes were regressed against dosage values of lead variants and the same covariates described for the main BOLT-LMM GWAS. The first 5 within-European principal components were also included as covariates to account for any subtle differences in ancestry. All exclusions and adjustments were made using baseline records (taken at the assessment centre).

**Mendelian randomisation (MR).** We performed two-sample MR, using the inverse-variance weighted (IVW) approach[75] as our main analysis method, and MR-Egger[75], weighted median estimation[76] and penalised weighted median estimation[76] as sensitivity analyses in the event of unidentified pleiotropy of our genetic instruments. Genetic variants that are robustly associated with the exposure of interest may also influence the outcome through associations with other risk factors for the outcome. This is known as "horizontal pleiotropy" and may bias MR results. The assumptions of IVW are that there is either no horizontal pleiotropy (under a fixed-effect model) or, if implemented under a random effects model due to heterogeneity among the causal estimates, that (i) there is no correlation between the strength of the association of the genetic instruments with the risk factor and the magnitude of the pleiotropic effects, and (ii) the pleiotropic effects have an average value of zero. Unbiased causal estimates can be obtained through MR-Egger if just the first condition above holds by estimating and adjusting for non-zero mean pleiotropy. If <50% of the weight in the analysis stems from variants that are pleiotropic then a weighted median approach may be used as an alternative. Given the differences in assumptions, if the results from all methods are broadly consistent, then our causal inference is strengthened.

In an effort to reduce the number of genetic instruments violating the above assumptions, we used a newly described method[77] to quantify, using a new iterative weighting method, each instrument's contribution to heterogeneity of the causal IVW estimate. High heterogeneity in Cochran's $Q$ statistic, which should follow a $\chi^2_{n-1}$ distribution for $n$ instruments, indicates that either invalid (horizontally pleiotropic) instruments have been included or that MR modelling assumptions have been violated. We therefore excluded variants with an extreme Cochran's $Q$ greater than the Bonferroni corrected threshold ($Q_{SNP} > \chi^2_{1-0.05/n,1}$) prior to performing MR analysis.

**Ethics and consent.** The UK Biobank was granted ethical approval by the North West Multi-centre Research Ethics Committee (MREC) to collect and distribute data and samples from the participants (http://www.ukbiobank.ac.uk/ethics/) and covers the work in this study, which was performed under UK Biobank application numbers 9072 and 16434. All participants included in these analyses gave informed consent to participate. UK Biobank consent procedures are detailed at http://biobank.ctsu.ox.ac.uk/crystal/field.cgi?id=200.

## Data availability

Summary statistics for the eight UK Biobank GWAS analyses can be found at http://www.t2diabetesgenes.org/data/ and on the Sleep Disorder Knowledge Portal at http://sleepdisordergenetics.org/informational/data/. The GGIR R script used to generate the activity-monitor measures is provided in Supplementary Data 10 and isavailable with the online version of this article.

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

## Acknowledgements

This research has been conducted using the UK Biobank Resource (applications 9072 and 16434). S.E.J. is funded by the Medical Research Council (grant: MR/M005070/1). S. Sabia is supported by EC Horizon2020 (LIFEPATH 633666). J.T. is funded by a Diabetes Research and Wellness Foundation Fellowship. M.A.T., M.N.W. and A.M. are supported by the Wellcome Trust Institutional Strategic Support Award (WT097835MF). A.R.W., T.M.F and H.Y. are supported by the European Research Council grants: SZ-245 50371-GLUCOSEGENES-FP7-IDEAS-ERC and 323195. H.Y. is also funded by Diabetes UK RD Lawrence fellowship (grant: 17/0005594). R.N.B. and R.M.F. are funded by the Wellcome Trust and Royal Society, grant 104150/Z/14/Z. K.S.R. is supported by funding from the Gillings Family Foundation. D.R.M. and P. G. are funded by the NIH (1R01HL143790-01). The generation and management of GWAS genotype data for the Rotterdam Study (RS-I, RS-II, RS-III) was executed by the Human Genotyping Facility of the Genetic Laboratory of the Department of Internal Medicine, Erasmus MC, Rotterdam, The Netherlands. The GWAS datasets are supported by the Netherlands Organisation of Scientific Research NWO Investments (nr. 175.010.2005.011, 911–03–012), the Genetic Laboratory of the Department of Internal Medicine, Erasmus MC, the Research Institute for Diseases in the Elderly (014–93–015; RIDE2), the Netherlands Genomics Initiative (NGI)/Netherlands Organisation for Scientific Research (NWO) Netherlands Consortium for Healthy Aging (NCHA), project nr. 050–060–810. The Rotterdam Study is funded by Erasmus Medical Centre and Erasmus University, Rotterdam, Netherlands Organisation for the Health Research and Development (ZonMw), the Research Institute for Diseases in the Elderly (RIDE), the Ministry of Education, Culture and Science, the Ministry for Health, Welfare and Sports, the European Commission (DG XII), and the Municipality of Rotterdam. The Rotterdam Study has been approved by the Medical Ethics Committee of the Erasmus MC (registration number MEC 02.1015) and by the Dutch Ministry of Health, Welfare and Sport (Population Screening Act WBO, license number 1071272–159521-PG). The Rotterdam Study has been entered into the Netherlands National Trial Register (NTR; www.trialregister.nl) and into the WHO International Clinical Trials Registry Platform (ICTRP, www.who.int/ictrp/network/primary/en/) under shared catalogue number NTR6831. The CoLaus study was and is supported by research grants from GlaxoSmithKline, the Faculty of Biology and Medicine of Lausanne, and the Swiss National Science Foundation (grants 33CSCO-122661, 33CS30–139468 and 33CS30–148401). The institutional Ethics Committee of the University of Lausanne, which afterwards became the Ethics Commission of Canton Vaud (www.cer-vd.ch) approved the baseline CoLaus study (reference 16/03, decisions of 13th January and 10th February 2003). The approval was renewed for the first (reference 33/09, decision of 23rd February 2009), the second (reference 26/14, decision of 11th March 2014) and the third (reference PB_2018–00040, decision of 20th March 2018) follow-ups. The Whitehall II study is supported by grants from the US National Institutes on Aging (R01AG013196; R01AG034454), the UK Medical Research Council (MRC K013351; MR/R024227/1), and British Heart Foundation (RG/13/2/30098). The University College London Hospital Committee on the Ethics of Human Research approved the study (reference number 85/0938). All participants provided written informed consent to participate in the study and to have their information obtained from treating physicians.

## Author contributions

The study was designed by T.M.F., P.G., V.T.v.H., S.E.J., D.R.M., M.N.W. and A.R.W. Participation in acquisition, analysis and/or interpretation of data was performed by N.A., R.N.B., H.S.D., J.E., T.M.F., R.M.F., J.W.H., V.T.v.H., Y.J., S.E.J., D.K., M.Z., Z.K., A.I.L., D.R.M., A.M., K.S.R., S.Sabia, R.S., S.Sharp, A.v.dS., H.T., M.A.T., J.T., P.M.V., M.N.W., A.R.W. and H.Y. Main writing group comprised of H.S.D. T.M.F., P.G., V.T.v.H, S.E.J., D.K., J.M.L., A.I.L. D.R.M., S.Sabia, H.T., M.K.T., P.M.V., M.N.W. and A.R.W. All authors reviewed this manuscript. A.R.W. is the guarantor of this work and, as such, had full access to all the data in the study and takes responsibility for the integrity of the data and the accuracy of the data analysis.

## Additional information

**Competing interests:** M.K.R reports receiving research funding from Novo Nordisk, consultancy fees from Novo Nordisk and Roche Diabetes Care, and modest owning of shares in GlaxoSmithKline. P.G. receives grant support from Merck, Inc. The other authors declare no competing interests.

Samuel E. Jones [1], Vincent T. van Hees [2], Diego R. Mazzotti[3,4], Pedro Marques-Vidal [5], Séverine Sabia[6,7], Ashley van der Spek[8], Hassan S. Dashti [9,10], Jorgen Engmann[11], Desana Kocevska[8,12], Jessica Tyrrell [1], Robin N. Beaumont [1], Melvyn Hillsdon[13], Katherine S. Ruth [1], Marcus A. Tuke [1], Hanieh Yaghootkar[1], Seth A. Sharp[1], Yingjie Ji[1], Jamie W. Harrison [1], Rachel M. Freathy [1], Anna Murray[1], Annemarie I. Luik[8], Najaf Amin[8], Jacqueline M. Lane[9,10], Richa Saxena[9,14,15], Martin K. Rutter[16,17], Henning Tiemeier[8,18], Zoltán Kutalik [19,20], Meena Kumari[21], Timothy M. Frayling[1], Michael N. Weedon[1], Philip R. Gehrman[3,4] & Andrew R. Wood[1]

[1]Genetics of Complex Traits, College of Medicine and Health, University of Exeter, Exeter EX2 5DW, UK. [2]Netherlands eScience Center, Amsterdam 1098 XG, The Netherlands. [3]Center for Sleep and Circadian Neurobiology, University of Pennsylvania, Philadelphia 19104 PA, USA. [4]Perelman School of Medicine of the University of Pennsylvania, Philadelphia 19104 PA, USA. [5]Department of Medicine, Internal Medicine, Lausanne University Hospital, Lausanne 1011, Switzerland. [6]Research Department of Epidemiology and Public Health, University College London, London WC1E 6BT, UK. [7]INSERM, U1153, Epidemiology of Ageing and Neurodegenerative diseases, Université de Paris, Paris 75010, France. [8]Department of Epidemiology, Erasmus MC University Medical Center, Rotterdam 3000 CA, The Netherlands. [9]Center for Genomic Medicine, Massachusetts General Hospital, Boston, MA 02114, USA. [10]Broad Institute of MIT and Harvard, Cambridge, MA 02142, USA. [11]UCL Institute of Cardiovascular Science, Research department of Population Science and Experimental Medicine, Centre for Translational Genomics, 222 Euston Road, London NW1 2DA, UK. [12]Department of Child and Adolescent Psychiatry, Erasmus Medical Center, Rotterdam 3000 CA, The Netherlands. [13]Sport and Health Sciences, College of Life and Environmental Sciences, University of Exeter, Exeter EX1 2LU, UK. [14]Department of Anesthesia, Critical Care and Pain Medicine, Massachusetts General Hospital and Harvard Medical School, Boston, MA 02111, USA. [15]Departments of Medicine, Brigham and Women's Hospital and Beth Israel Deaconess Medical Center, Harvard Medical School, Boston, MA 02115, USA. [16]Division of Diabetes, Endocrinology and Gastroenterology, Faculty of Medicine, Biology and Health, University of Manchester, Manchester M13 9PL, UK. [17]Manchester Diabetes Centre, Manchester University NHS Foundation Trust, Manchester Academic Health Science Centre, Oxford Road, 193 Hathersage Road, Manchester M13 0JE, UK. [18]Department of Social and Behavioral Science, Harvard TH Chan School of Public Health, Boston, MA 02115, USA. [19]Institute of Social and Preventive Medicine (IUMSP), Lausanne University Hospital, Lausanne 1010, Switzerland. [20]Swiss Institute of Bioinformatics, Lausanne 1015, Switzerland. [21]ISER, University of Essex, Colchester, Essex CO4 3SQ, UK. These authors contributed equally: Samuel E. Jones, Vincent T. van Hees, Diego R. Mazzotti. These authors jointly supervised this work: Timothy M. Frayling, Michael N. Weedon, Philip Gehrman, Andrew R. Wood

