## [peer review file · Nature Communications]

Reviewer #1 (Remarks to the Author):

In this manuscript, the authors report GWAS of accelerometer-derived phenotypes in UK biobank. The manuscript is well written, and the authors generally follow the current standards (and the set of tools used) in GWAS papers. I provide comments here mostly ordered by corresponding sections in the manuscript.

1. Results, first paragraph. It's preferable to use the term "phenotypic correlation", over "observational correlation".
2. Results, section titled "Forty-seven genetic association...". LD-score intercepts ranged from 1.03 to 1.07, while lambda GC was up to 1.14. $1.14 > 1.07$. How do you conclude that "any inflation of test statistics observed is more likely due to the polygenicity of the phenotype"?
3. Results, section titled "Replication of 47 genetic associations...". Can you add how many associations replicated individually?
4. Same section. "For traits with more than one SNP associated at $P < 5 \times 10^{-8}$... we combined the effects of each SNP" -- was this performed for lead SNPs from independent genomic regions?
5. Results, section "Variants associated with sleep quality...". Are "sleep quality" and "sleep efficiency" interchangeable? Can you clarify whether you use sleep efficiency measures to reflect sleep quality, or you view both as the same?
6. Results, section "Fine-mapping analysis identifies multiple...". Please present the Bayes factor condition after you say you're using FINEMAP, because outside this software, there is no general rule of using Bayes factors with this condition to identify variant likely to be causal.
7. Same section. The term "genomic region" is better than "locus", because locus can refer to a single variant.
8. Same section. When you refer to the missense APOE variant "representing" the e4 allele, do you mean that it is the e4 allele? or that it is a proxy?
9. Same section, which tissue did you use from GTEx?
10. Results, section "Multiple sleep traits have genetic variants....". Please update the title to "Multiple sleep traits have associated genetic variants...".
11. Same section. MR analysis assumes that the SNPs are associated with the sleep traits only via their effect on restless leg syndrome. Can you be certain this holds? if so (and if you're not), please state it, and explain why. Also, the conclusions of variants associated with restless leg syndrome not being artifacts of accelerometer is not clear, given that the restless leg syndrome variants were from a different GWAS, please clarify.
12. Results, section "Waist-hip-ratio)... ". The first sentence is: "Given genetic correlations are...". I'm not sure this sentence is true, despite cited paper. I suggest writing instead "Because xxx reported that genetic correlations are generally..." (Also: observational correlations = phenotypic correlations?).

13. Same section. Using genetic correlations to prioritize results for MR analysis is dubious. While I realize this is becoming common in the field, I believe this is unfounded. MR analysis assumes that the genetic variants used are directly, causally, associated with one trait (for which we study causality), and are not directly associated with the outcome trait. This is a very strong assumption, and it is much more likely that relationship between traits are synchronous. Further, taking the traits that have high genetic correlation potentially only enhances this problem. Finally, bi-directional MR may be heavily influenced by the sample size used to generate each of the GWAS for the two traits. More practically, my recommendation is to remove the first sentence of this paragraph, clarify the assumptions of MR, and state the caveats behind your conclusions.

14. Statistical analysis: did anyone report that using GRM estimated in BOLT-LMM in your data suffices for population structure and relatedness adjustment? how do you know that is suffices? especially given that your analysis was inflated. Please report how you made sure that population structure is sufficiently controlled, or alternatively, someone else's work that demonstrated that it is sufficiently controlled (e.g. simulation studies showing that tests of genetic variants that are not associated with the outcome have proper distribution). Also, in sensitivity analyses you adjusted for 5 PCs. How come you did not adjust for 5 PCs in the main analysis?

15. Statistical analysis, fine mapping. Is the "index variant" the "lead variant"? or please define this.

16. Statistical analysis, fine mapping. "full UK Biobank imputed genotype probabilities" -- you mean expectations? (dosages are expectations, not probabilities, so unless you imputed phased chromosomes, you likely mean dosages/expectations).

17. Table 1: what is "Pseudo heritability"? I believe that what one tries to estimate is heritability.

18. Figure 2: you can use "effect estimates" rather than "betas" (this is true for other figures as well). The sentence "Betas represent standard deviations..." should be made clearer, e.g. "Effects sizes are given as the per-allele change in units of standard deviation of the normalized trait".

19. Figure 2: Did your analysis used inverse normal transformation? did was not reported earlier. What about the replication studies?

20. Methods, replication studies: you did not report covariates, methods.

21. Figure 4: I don't think these report correlations, as the caption suggests.

22. The last sentence of the discussion, pointing at new therapeutic targets seems like a big overstatement.

23. The authors write that detected variants are enriched for serotonin precessing genes and cerebral expressed genes, and that this provides new biological insights into sleep characteristics. While there is a short paragraph about serotonin in the discussion, can you also add a review of sleep (non genetics) literature relation to cerebral control of sleep (say)? I wonder if these insights are really new, or a nice confirmation of existing hypotheses.

Reviewer #2 (Remarks to the Author):

In this interesting report, a GWAS is conducted on actigraphy measures in 85,000 people with replication in 5,900 people, identifying 47 sleep trait associations. It is a very interesting report with valid findings. The analysis is done stringently, and there is even an independent replication dataset.

My major issue is that the study is not critical enough on what activity measures can and cannot deliver; “actigraphy-derived sleep traits” are taken at face value. Even so it is certainly better than self-report, min per min actigraphy cannot differentiate awake and not moving from asleep so it is only a proxy. The authors need to discuss the weakness of the approach much more thoroughly, and I believe their interpretation are overreaching at this state of our understanding.

For example, actigraphy derived measures of sleep duration of this study are associated both with Pax8 (loci associated with self-reported sleep duration but not RLS) and MEIS1/BTBD9 (associated with RLS but not self-reported sleep duration). It is however really hard to exclude the possibility that PLMs or RLS is contaminating some of these findings, as suggested by El Gewely Sleep. 2018 Sep 12 regarding the role of MEIS1 on insomnia; the study found that MEIS has effects on RLS but not on insomnia when the phenotypes are carefully screened out. This paper also has a nice discussion in the introduction on the pro and cons of considering the UK biobank MEIS association with insomnia partially or totally confounded by RLS. My issue with the present report is that in the discussion the authors reject the possibility that PLMs are confounding without data to support it.

Specifically:

Abstract: I would add a sentence in the abstract outline the limitation of this technology, for example: Although actigraphy-derived measures are imperfect measures of true sleep and sleep disorders, these findings provide new biological insights into sleep characteristics in comparison to other genetic studies that have only used subjective reports.

Introduction: “Research-grade activity monitors (accelerometers), also known as actigraphy devices, provide cost-effective estimates of sleep using validated algorithms^{24,25}.”

The term “validation” is often used as a magic wand, but it is only valid in the context it has been validated in. To my knowledge, actigraphy measures have not been validated in large population-based samples, and it is likely not to measure well or proportionally well a number of the derived features it is claimed to do if patients have sleep apnea, insomnia, parasomnias, medical disorders, a large part of the population studied. Naps are not well captured with actigraphy, and triaxial accelerometers generally have the problem of either being calibrated to measure well intense movements for exercise, or smaller movement for sleep, not both.

This is reflected by very poor choice of references used for this “validation” statement. Reference 24 only studied 24 subjects including 4 only severe sleep apnea and 2 insomnia patients for example, not allowing clear conclusions on group effects that could disrupt linearity of accelerometer-based measures versus true sleep measures across pathologies. Reference 26 is not a validation at all, but a use-case showing heuristic value in predicting mortality. There are many references comparing PSG and actigraphy, and they all show problems as they can be calibrated to detect well sleep but then not wake, or vice versa (see for example the reference Pigeon et al., *J Clin Sleep Med*. 2018 Jun 15;14(6):1057-1062, which has among the best performance but still show PPV of ~70% in normal sleepers.

The authors show association with restless legs syndrome-RLS (these patients also have PLMs), insomnia and chronotype genes in all the different phenotypes measured by actigraphy, and the authors are arguing these effects are all direct on the “sleep” derived phenotype. These phenotypes are indirect, complex and interact in many ways, and it is somewhat a leap of faith.

For example, the authors used Mendelian randomization to show a causative association of RLS syndrome with actigraphy based sleep duration and chronotype and self-reported sleep duration and chronotype. I am thus not sure why the authors then suggest in the discussion that this excludes measurement issues due to PLMS for example, since almost all RLS subjects have PLMS. I quote: “A subset of variants previously associated with restless legs syndrome were associated with sleep duration, quality and timing measures. This observation is unlikely to be an artefact caused by limb movements during sleep because we found that the same variants are associated with self-report measures of sleep duration, chronotype and insomnia.” It is well known that both PLMs and RLS have circadian effects (Michaud et al., *Ann Neurol*. 2004 Mar;55(3):372-80) which is one way this could influence circadian phenotypes, plus see above for confounding effects of RLS on insomnia. Finally, it would be logical that if a subject moves his legs every 30 sec without waking up (a common feature of PLMs), actigraphy-based measures of immobility are going to be affected. PLMs are extremely common beyond insomnia and RLS. For example 40% of older subjects have PLMI >15/hr.

In brief, self-reported measures of sleep duration could be biased in the same direction as the accelerometer-based measures of sleep duration, but for different reasons, and not related to actual total sleep time if it was measured by EEG. I am not trying to say that this paper is not a laudable effort in the direction of more objective measurements versus self-reports, but it is trying to say these measures are good through very indirect reasoning. Although it is my opinion these RLS genes have pleiotropic effects on the motor system and sleep itself, I would be more cautious until we know more and this has been formally tested, especially considering the fact associations with PLMs have not yet been reported in large samples.

It is a detail, but I am not so sure the paragraph: “Melatonin is frequently taken as a dietary supplement in the United States with its use more than doubling between 2007 and 2012⁴⁸, although clinical trial results for sleep and circadian rhythm disorders are mixed⁴⁹. In addition, excess melatonin levels can also lead to disturbed sleeping and other health issues with the American Academy of Sleep Medicine recommending avoiding melatonin for chronic insomnia⁵⁰.” is really relevant to this publication.

The rest of the findings are relevant and interesting.

Suppl Table 1 is really essential to the interpretation of this data and should be a main table.

Reviewer #3 (Remarks to the Author):

This is the latest in a series of manuscripts to identify novel loci for sleep-related phenotypes in large samples. While other studies (Dashti et al., BioRxiv 2018, Lane et al. BioRxiv 2018, Jansen et al. BioRxiv 2018) have been based upon self-report rather than actigraphy (accelerometer-derived data), support for the loci identified in two of those studies has been sought in the actigraphy data analysed here.

The authors identify 47 genetic associations surpassing the standard genome-wide threshold of significance ($P < 5 \times 10^{-8}$). In the replication sample, direction of effect is convincingly replicated for the top 20 associations with $P < 8 \times 10^{-10}$, but much less so for the remaining 27. When reporting on individual loci, it would be useful to indicate to which of these two groups they belong.

A brief mention of the potential for ascertainment/survival bias would be useful when first presenting the APOE association, rather than leaving it to the final discussion (the description of this in the discussion section could also be a bit clearer).

The conclusion that lower genetic correlation between self-report and actigraphy-measured sleep duration ‘... suggests differences in the genetic contribution to variation in self-reported versus objective sleep duration’ seems a bit strange. Surely both measures seek to capture the same trait, the difference being that actigraphy is presumably the more accurate of the two. Both actigraphy and self-report measures of sleep duration appear to have been analysed here on the same set of ~85,670 individuals (the methods are not completely clear on this point). Is the higher correlation between actigraphy-measured sleep and activity timings and self-report chronotype possibly due to

chronotype being analysed in a much larger study of 449,734 individuals? If self-report chronotype is analysed in the same set of individuals as the actigraphy data, do self-report & objective measures still show much greater correlation than for sleep duration?

The description of the gene-set enrichment analysis performed using MAGMA is minimal. How was Bonferroni correction performed – separately for each trait? How many gene sets were tested (& from which sources)? What settings were used in MAGMA – was a window used to assign nearby SNPs to genes; which gene analysis model was used; was the gene-set test competitive or self-contained...? This information should be presented in the methods. Despite all sleep traits presumably being tested, only one finding is reported (enrichment of serotonin metabolism genes for association with number of nocturnal sleep episodes). Is this the only enrichment that survives correction for this trait? Are there no associations for any other trait? All associations surviving correction for each trait (if any) should at least be presented in a supplementary table.

Dear Michelle,

Thank you for considering our manuscript. We have addressed all of the reviewer's comments below. We think the manuscript is much improved.

Reviewer #1 (Remarks to the Author):

In this manuscript, the authors report GWAS of accelerometer-derived phenotypes in UK biobank. The manuscript is well written, and the authors generally follow the current standards (and the set of tools used) in GWAS papers. I provide comments here mostly ordered by corresponding sections in the manuscript.

We thank the Reviewer for their positive comments.

1. Results, first paragraph. It's preferable to use the term "phenotypic correlation", over "observational correlation".

We agree with the reviewer and have modified the text accordingly in this section (page 5) and in the third sentence of "Waist-hip-ratio (adjusted for BMI) and educational attainment causally influence sleep outcomes" section (page 11) to be consistent throughout.

2. Results, section titled "Forty-seven genetic association...". LD-score intercepts ranged from 1.03 to 1.07, while lambda GC was up to 1.14. $1.14 > 1.07$. How do you conclude that "any inflation of test statistics observed is more likely due to the polygenicity of the phenotype"?

We take the reviewer's point and have changed the wording of this sentence so we do not completely rule out this possibility and instead suggest this is unlikely because 1) the relatively low values of the LD-score intercept and 2) how the LD-score intercept can be affected by sample size and heritability (Loh et al. 2018 PMID: 29892013). More generally, for sample sizes as large those presented in this analysis, the LD-score intercept is more critical than lambda GC because it accounts for polygenicity which can inflate the median chi-square statistic as sample sizes increase (Bulik-Sullivan et al. 2015 PMID: 25642630). Previous simulations of quantitative traits in the absence of confounding have shown LD-score intercepts < 1.1 to be suggestive of minimal confounding (Bulik-Sullivan et al. 2015 PMID: 25642630).

3. Results, section titled "Replication of 47 genetic associations...". Can you add how many associations replicated individually?

This is a good point from the reviewer – the availability of such a large sample size 85,000 means our discovery dwarfs our replication data and replication of individual variants is not well powered. We have updated this section to provide more detail as follows (page 7):

"Of the 47 associations, the signal near GPR139 (rs8045740) reached Bonferroni significance ($P=0.001$) and 11 were associated at $P<0.05$ after meta-analysis of the replication studies. Given the limited power to detect single SNP associations in the replication meta-analysis, we next examined the directional consistency of allele effect estimates."

4. Same section. "For traits with more than one SNP associated at $P < 5 \times 10^{-8}$... we combined the effects of each SNP" -- was this performed for lead SNPs from independent genomic regions?

The analyses used independent lead SNPs as outlined in Table 3 (previously Table 2). We have updated the text in the manuscript to make this clearer (page 7).

5. Results, section "Variants associated with sleep quality...". Are "sleep quality" and "sleep efficiency" interchangeable? Can you clarify whether you use sleep efficiency measures to reflect sleep quality, or you view both as the same?

In the last paragraph of the introduction we defined sleep quality as a term that encompasses sleep efficiency and the number of nocturnal sleep episodes and have updated to clarify as follows (page 5):

"These included measures representative of sleep quality, including sleep efficiency (sleep duration divided by the time between the start and end of the first and last nocturnal inactivity period, respectively) and the number of nocturnal sleep episodes."

We have also amended the first sentence to clarify this at the start of the "Variants associated with sleep quality..." section (page 8):

"Of the 5 variants associated with sleep efficiency, a measure of sleep quality, ..."

6. Results, section "Fine-mapping analysis identifies multiple...". Please present the Bayes factor condition after you say you're using FINEMAP, because outside this software, there is no general rule of using Bayes factors with this condition to identify variant likely to be causal.

We have modified the text at the start of this section as follows (page 10):

"To identify credible SNP sets likely to contain causal variants within 500Kb of lead SNPs for each trait with a genetic association ($P < 5 \times 10^{-8}$) we used FINEMAP⁴¹ to identify credible sets of likely causal SNPs (\log_{10} Bayes Factor > 2) (Supplementary Table 7). This approach places a probability on the likelihood that a variant, amongst those tested, represents the causal allele."

7. Same section. The term "genomic region" is better than "locus", because locus can refer to a single variant.

We thank the reviewer for this suggestion but feel that "locus" is standard nomenclature in genome-wide association publications when used to refer to a "genomic region".

8. Same section. When you refer to the missense AOPE variant "representing" the e4 allele, do you mean that it is the e4 allele? or that it is a proxy?

We agree with the reviewer that we needed to clarify this. We mean proxy in the same way as described in the "Variants associated with sleep quality include known restless legs syndrome, sleep duration, and cognitive decline associated variants" section. We have changed the text to clarify this point as follows (page 10):

“The other was the missense APOE variant, a proxy for the $\epsilon 4$ allele known to predispose to Alzheimer’s disease and responsible for the association signal with the number of nocturnal sleep episodes.”

9. Same section, which tissue did you use from GTEx?

We looked up all “plausible” candidate variants available from the GTEx consortium to determine whether the candidate SNP (or proxy) was the strongest eQTL across ALL tissues by using results of an all-tissue meta-analysis of variant-gene expression levels. As our phenotypes are behavioural in nature, we had performed (but not included for brevity) analyses using only results from brain tissues – these are now included in the relevant table (Supplementary Table 7). If our fine-mapped variant represents an eQTL, we report each gene and the corresponding tissue with strongest evidence of association with that gene’s expression levels. The Methods section (“Alamut annotation and eQTL mapping” subsection) has been updated to reflect the inclusion of these additional results as follows (page 22):

“To investigate whether the fine-mapped SNPs were eQTLs, we searched for our SNPs in the single-tissue cis-eQTL and multi-tissue eQTL datasets (v7), available at the GTEx portal (<https://www.gtexportal.org/home/datasets>) for significant SNP-gene eQTL associations. In the multi-tissue eQTL data, we reported a SNP as an eQTL for a gene if the SNP-gene association was significant in the meta-analysis across all tissues. Using the single-tissue cis-eQTL data, we performed a lookup of our fine-mapped SNPs for significant SNP-gene associations in brain tissues only. For each gene with a fine-mapped SNP significantly associated with expression levels, we reported the tissue with the strongest evidence (lowest P-value) of association and the correlation (r^2) between the fine-mapped SNP and the tissue’s strongest eQTL SNP.”

10. Results, section "Multiple sleep traits have genetic variants....". Please update the title to "Multiple sleep traits have associated genetic variants...".

We have updated the title of this section to “Genetic variants known to be associated with restless legs syndrome are associated with multiple sleep traits” (page 10).

11. Same section. MR analysis assumes that the SNPs are associated with the sleep traits only via their effect on restless leg syndrome. Can you be certain this holds? if so (and if you're not), please state it, and explain why. Also, the conclusions of variants associated with restless leg syndrome not being artifacts of accelerometer is not clear, given that the restless leg syndrome variants were from a different GWAS, please clarify.

We can’t be certain that the SNPs act through their effect on RLS; it could be due to periodic limb movement as discussed by Reviewer 2 - comment 1, or some other mechanism. We think the fact the RLS MR also demonstrates causal association with self-report measures of sleep strongly supports our argument that it is unlikely to be just due to artefacts of limb movement from RLS influencing our activity monitor derived measures of sleep. We have also performed multiple MR sensitivity analyses that account for potential issues such as pleiotropy - all are consistent with a causal effect of RLS on sleep measures. Please also see our response to Reviewer 2 - comment 1.

12. Results, section "Waist-hip-ratio)... ". The first sentence is: "Given genetic correlations are...". I'm not sure this sentence is true, despite cited paper. I suggest writing instead "Because xxx reported that genetic correlations are generally..." (Also: observational correlations = phenotypic correlations?).

We have modified the text to incorporate this suggestion and have also replaced "observational" with "phenotypic" (page 11).

13. Same section. Using genetic correlations to prioritize results for MR analysis is dubious. While I realize this is becoming common in the field, I believe this is unfounded. MR analysis assumes that the genetic variants used are directly, causally, associated with one trait (for which we study causality), and are not directly associated with the outcome trait. This is a very strong assumption, and it is much more likely that relationship between traits are synchronous. Further, taking the traits that have high genetic correlation potentially only enhances this problem. Finally, bi-directional MR may be heavily influenced by the sample size used to generate each of the GWAS for the two traits. More practically, my recommendation is to remove the first sentence of this paragraph, clarify the assumptions of MR, and state the caveats behind your conclusions.

We agree with the reviewer in that we have not necessarily made it clear why we have relied on genetic correlations to guide our MR analyses. As genetic correlations can be derived using just the publicly-available summary statistics, we can calculate genetic correlations (and infer phenotypic correlations) for phenotypes that are a) under-represented, b) not recorded and c) not well-defined in our dataset (i.e. the UK Biobank). To remain consistent, we used genetic correlations to guide MR analyses instead of a combination of genetic and phenotypic correlations. Further justification is that we are interested in assessing causality of phenotypes where there is some evidence of genetic overlap, as these are the phenotypes for which we will have statistical power to assess direction of association through Mendelian Randomisation. We have added this explanation to the beginning of this results section (page 11), replacing:

"Given genetic correlations are generally similar to observational correlations⁴¹, we used genetic correlations to prioritise traits for subsequent Mendelian Randomisation analyses."

with

"To assess causality of phenotypes, we used genetic correlations to prioritise traits with evidence of genetic overlap for subsequent Mendelian Randomisation analyses using LD-Hub³². We tested for genetic correlations between the 8 activity monitor derived measures and 234 published GWAS studies across a range of diseases and traits. Given previous reports that genetic correlations are similar to phenotypic correlations⁴⁴, this approach also enabled us to analyse phenotypes under-represented, not recorded, or not well defined within the UK Biobank."

14. Statistical analysis: did anyone report that using GRM estimated in BOLT-LMM in your data suffices for population structure and relatedness adjustment? how do you know that is suffices? especially given that your analysis was inflated. Please report how you made sure that population structure is sufficiently controlled, or alternatively, someone else's work that

demonstrated that it is sufficiently controlled (e.g. simulation studies showing that tests of genetic variants that are not associated with the outcome have proper distribution). Also, in sensitivity analyses you adjusted for 5 PCs. How come you did not adjust for 5 PCs in the main analysis?

Linear-mixed models (LMMs), including those as implemented in BOLT-LMM, have been shown to be an effective way to account for stratification and relatedness through the estimation of “genomic relatedness” (Kang et al. 2010 pubmed ID: 20208533, Zhou et al. 2012 pubmed ID: 22706312, Yang et al. 2014 pubmed ID: 24473328, Loh et al. 2015 pubmed ID: 25642633, Loh et al. 2018 pubmed ID: 29892013).

The random effects estimated by the LMM incorporate structure from the samples in the analysis whereby the components of the structure may represent differences in ancestry, relatedness, or both. Additional adjustment for principal components in these models would result in an over correction for differences in ancestry and so we have not including principal components as additional covariates in the main LMM-based analysis. However, in the analyses outside of the LMM framework that incorporates unrelated individuals only (as estimates through kinship coefficients), we have included the first 5 principal components as generated within the set of Europeans we had defined. We have added an explanation to clarify this point in the “Sensitivity Analysis” subsection of the Methods by adding the following (page 23):

“Relatedness was inferred using kinship coefficients provided by the UK Biobank. The maximal set of unrelated individuals (3^{rd} degree) were put forward for sensitivity analyses.”

and

“Sensitivity analyses were performed using fixed-effect models. Phenotypes were regressed against dosage values of lead variants and the same covariates described for the main BOLT-LMM GWAS. The first 5 “within-European” principal components were also included as covariates to account for any subtle differences in ancestry.”

15. Statistical analysis, fine mapping. Is the "index variant" the "lead variant"? or please define this.

The index variant is indeed the lead variant. As this was not clear, we have replaced all instances of the phrase “index variant” with “lead variant”.

16. Statistical analysis, fine mapping. "full UK Biobank imputed genotype probabilities" -- you mean expectations? (dosages are expectations, not probabilities, so unless you imputed phased chromosomes, you likely mean dosages/expectations).

We used dosages for this analysis – but we derived dosage values using the genotype probability data encoded within the “bgen” data format: (http://www.well.ox.ac.uk/~qav/bgen_format/bgen_format_v1.2.html). We have made a subtle edit to this section to make it clearer (page 21):

“Dosages were derived from the unrelated European subset of the full UK Biobank imputed genotype probabilities (N=379,769).”

17. Table 1: what is "Pseudo heritability"? I believe that what one tries to estimate is heritability.

The term "pseudo heritability" originated from Kang et al 2010 pubmed ID: 2020853. This represents the variance explained by a relatedness matrix derived from the use of SNP-chips rather than the use of a pedigree. However, to avoid confusion we have removed "pseudo" from the manuscript. Please note also that the table referred to is now Table 2 following suggestion from Reviewer 2 - comment 6.

18. Figure 2: you can use "effect estimates" rather than "betas" (this is true for other figures as well). The sentence "Betas represent standard deviations..." should be made clearer, e.g. "Effects sizes are given as the per-allele change in units of standard deviation of the normalized trait".

We thank the reviewer for this suggestion. We have now updated the term for all figure legends.

19. Figure 2: Did your analysis used inverse normal transformation? did was not reported earlier. What about the replication studies?

Accelerometer-derived phenotypes were analyzed both on the original scale and inverse-normal scale. We have based our GWAS discovery on the inverse-normal scale to reduce the number of false positives that may arise with rare/low-frequency variants due to the skewness of some the distributions. However, we have used the effect estimates from the raw scale in minutes to provide context in the following sections: "Variants associated with sleep quality include known restless legs syndrome, sleep duration, and cognitive decline associated variants" and "Ten novel sleep duration loci identified from accelerometer-derived sleep duration GWAS". As part of the "Statistical Analysis" methods section, we have updated the "Genome-wide association analyses" sub-section (page 20) as follows:

"Phenotypes were analysed on their original-scale and the inverse-normal-scale after transformation and all results (except those that refer to interpretable effect sizes) are reported for the inverse-normal scale analyses."

We have also updated the "Replication of findings" sub-section (page 22) as follows:

"For all replication studies, the derivation of the sleep characteristics and the same overall- and trait-specific exclusion criteria outlined above applied. Where available, accelerometer-derived phenotypes were analyzed both on the original scale and inverse-normal scale."

20. Methods, replication studies: you did not report covariates, methods.

For all replication studies, the covariates incorporated into the model were the same as those used in the UK Biobank analysis (outlined in the methods). No study-specific covariates were used. We have incorporated this information into the methods section detailing the replication effort as follows (page 22):

"The covariates incorporated into the model were the same as those used in the UK Biobank analysis."

21. Figure 4: I don't think these report correlations, as the caption suggests.

We thank the reviewer for noting this – we have changed the beginning of the legend to “Comparisons”.

22. The last sentence of the discussion, pointing at new therapeutic targets seems like a big overstatement.

We have removed this statement from the discussion (page 16).

23. The authors write that detected variants are enriched for serotonin precessing genes and cerebral expressed genes, and that this provides new biological insights into sleep characteristics. While there is a short paragraph about serotonin in the discussion, can you also add a review of sleep (non genetics) literature relation to cerebral control of sleep (say)? I wonder if these insights are really new, or a nice confirmation of existing hypotheses.

We thank the reviewer for their suggestion and have now added reference to the review by Brown et al 2012 focussing on the control of sleep and wakefulness through mechanisms in the brain. The finding that loci were enriched for genes in the serotonin signalling pathways coincide with what is already known about the pathways that control sleep-wake cycles. The serotonin system is part of the ascending reticular activating system that, among other things, contributes to the regulation of sleep and wake states. As such, it is not surprising (though indeed a useful confirmation) that genetic variation in the serotonin signalling pathway would be implicated in sleep-related traits. This is consistent with past candidate gene studies that have found an association between the serotonin transporter gene and insomnia (e.g. pubmed ID: 20337192).

Reviewer #2 (Remarks to the Author):

In this interesting report, a GWAS is conducted on actigraphy measures in 85,000 people with replication in 5,900 people, identifying 47 sleep trait associations. It is a very interesting report with valid findings. The analysis is done stringently, and there is even an independent replication dataset.

We thank the reviewer for their comments.

1. My major issue is that the study is not critical enough on what activity measures can and cannot deliver; “actigraphy-derived sleep traits” are taken at face value. Even so it is certainly better than self-report, min per min actigraphy cannot differentiate awake and not moving from asleep so it is only a proxy. The authors need to discuss the weakness of the approach much more thoroughly, and I believe their interpretation are overreaching at this state of our understanding.

For example, actigraphy derived measures of sleep duration of this study are associated both with Pax8 (loci associated with self-reported sleep duration but not RLS) and MEIS1/BTBD9 (associated with RLS but not self-reported sleep duration). It is however really hard to exclude the possibility that PLMs or RLS is contaminating some of these findings, as suggested by El Gewely Sleep. 2018 Sep 12 regarding the role of MEIS1 on insomnia; the study found that MEIS1 has effects on RLS but not on insomnia when the phenotypes are carefully screened out. This paper also has a nice discussion in the introduction on the pro and cons of considering the UK biobank MEIS association with insomnia partially or totally confounded by RLS. My issue with the present report is that in the discussion the authors reject the possibility that PLMs are confounding without data to support it.

We agree that the limitations of accelerometry for differentiating sleep from lack of movement when awake was not adequately discussed. We have added a brief description of these issues in the limitations paragraph of the Discussion and also referred the reader to our recent paper (van Hees et al. Scientific Reports, 2018) that describes these issues in greater detail as follows (pages 15-16):

“Finally, it is important to keep in mind that while accelerometry provides a more objective means of assessing sleep and wake than self-report, it has its own limitations. For people with insomnia, accelerometry tends to overestimate sleep because time spent lying still in bed awake attempting to sleep can be scored as sleep⁵⁹. However, most studies have relied on a devices that measure a single axis of movement that could be more prone to these errors, and our recent work suggests that newer tri-axial devices may be more accurate³⁰.”

Please see our response to comment #4 below for how we address potential confounding of RLS and PLMS.

Specifically:

2. Abstract: I would add a sentence in the abstract outline the limitation of this technology, for example: Although actigraphy-derived measures are imperfect measures of true sleep and sleep disorders, these findings provide new biological insights into sleep characteristics in comparison to other genetic studies that have only used subjective reports.

We have amended the last sentence in the abstract to recognise the imperfect nature of accelerometers in deriving sleep characteristics:

Although accelerometer-derived measures of sleep are imperfect, these findings provide new biological insights into sleep characteristics in comparison to previous efforts based on subjective measures of sleep.

3. Introduction: “Research-grade activity monitors (accelerometers), also known as actigraphy devices, provide cost-effective estimates of sleep using validated algorithms^{24,25}.”

The term “validation” is often used as a magic wand, but it is only valid in the context it has been validated in. To my knowledge, actigraphy measures have not been validated in large population-based samples, and it is likely not to measure well or proportionally well a number of the derived features it is claimed to do if patients have sleep apnea, insomnia, parasomnias, medical disorders, a large part of the population studied. Naps are not well captured with actigraphy, and triaxial accelerometers generally have the problem of either being calibrated to measure well intense movements for exercise, or smaller movement for sleep, not both.

This is reflected by very poor choice of references used for this “validation” statement. Reference 24 only studied 24 subjects including 4 only severe sleep apnea and 2 insomnia patients for example, not allowing clear conclusions on group effects that could disrupt linearity of accelerometer-based measures versus true sleep measures across pathologies. Reference 26 is not a validation at all, but a use-case showing heuristic value in predicting mortality. There are many references comparing PSG and actigraphy, and they all show problems as they can be calibrated to detect well sleep but then not wake, or vice versa (see for example the reference Pigeon et al., J Clin Sleep Med. 2018 Jun 15;14(6):1057-1062, which has among the best performance but still show PPV of ~70% in normal sleepers.

We agree with the reviewer and have now re-phrased the term “validated” as: “an acceptable level of agreement with polysomnography and sleep diaries”. Note that in van Hees et al. 2018, the mean C-statistic to detect the sleep period time window compared to polysomnography was 0.86 and 0.83 in clinic-based (N=28) and healthy (N=22) sleepers, respectively. We have also removed references previously 24 and 25 from the manuscript and now refer to the following papers: van Hees et al, Plos One, 2015 & van Hees et al, Scientific Reports, 2018 (references 29 & 30 of the revised manuscript).

The statement by the reviewer that “triaxial accelerometers generally have the problem of either being calibrated to measure well intense movements for exercise, or smaller movement for sleep, not both” mainly applies to the conventional actigraphy devices. There is growing body of literature to support the use of raw data accelerometry for both sleep and physical activity research. It is not the number of axes as such, but the access to the raw data that facilitates tailored analysis on the same data for sleep and daytime activity.

Please note that our recent evaluation includes a comparison with PSG in sleep clinic patients. The study sample was small and did not allow us to make specific statement per sleep disorder.

However, results from the group as a whole indicated that the method has value for describing sleep in this population.

4. The authors show association with restless legs syndrome-RLS (these patients also have PLMs), insomnia and chronotype genes in all the different phenotypes measured by actigraphy, and the authors are arguing these effects are all direct on the “sleep” derived phenotype. These phenotypes are indirect, complex and interact in many ways, and it is somewhat a leap of faith.

For example, the authors used Mendelian randomization to show a causative association of RLS syndrome with actigraphy based sleep duration and chronotype and self-reported sleep duration and chronotype. I am thus not sure why the authors then suggest in the discussion that this excludes measurement issues due to PLMS for example, since almost all RLS subjects have PLMS. I quote: “A subset of variants previously associated with restless legs syndrome were associated with sleep duration, quality and timing measures. This observation is unlikely to be an artefact caused by limb movements during sleep because we found that the same variants are associated with self-report measures of sleep duration, chronotype and insomnia.” It is well known that both PLMs and RLS have circadian effects (Michaud et al., *Ann Neurol.* 2004 Mar;55(3):372-80) which is one way this could influence circadian phenotypes, plus see above for confounding effects of RLS on insomnia. Finally, it would be logical that if a subject moves his legs every 30 sec without waking up (a common feature of PLMs), actigraphy-based measures of immobility are going to be affected. PLMs are extremely common beyond insomnia and RLS. For example 40% of older subjects have PLMI >15/hr.

In brief, self-reported measures of sleep duration could be biased in the same direction as the accelerometer-based measures of sleep duration, but for different reasons, and not related to actual total sleep time if it was measured by EEG. I am not trying to say that this paper is not a laudable effort in the direction of more objective measurements versus self-reports, but it is trying to say these measures are good through very indirect reasoning. Although it is my opinion these RLS genes have pleiotropic effects on the motor system and sleep itself, I would be more cautious until we know more and this has been formally tested, especially considering the fact associations with PLMs have not yet been reported in large samples.

We agree that the potential influence of RLS/PLMS was downplayed too much and have integrated their potential confounding effects into the Discussion section as follows (page 13):

“Given that the same variants are also associated with self-report measures of sleep duration, chronotype and insomnia, this observation may not be an artefact caused by limb movements during sleep. On the other hand, the repetitive periodic limb movements (PLMS) that people with RLS typically experience during sleep could have been detected by the accelerometers and confounded the parameters. Studies with more in-depth phenotyping of sleep disorders are needed to more fully evaluate the contribution of RLS and PLMS to sleep traits, especially in light of a recent paper showing that associations with MEIS1 were only in those with RLS⁵³.”

5. It is a detail, but I am not so sure the paragraph: “Melatonin is frequently taken as a dietary supplement in the United States with its use more than doubling between 2007 and 2012⁴⁸, although clinical trial results for sleep and circadian rhythm disorders are mixed⁴⁹. In addition, excess melatonin levels can also lead to disturbed sleeping and other health issues with the

American Academy of Sleep Medicine recommending avoiding melatonin for chronic insomnia⁵⁰.” is really relevant to this publication.

We have removed this sentence and corresponding references from the manuscript.

The rest of the findings are relevant and interesting.

We thank the reviewer for their comment.

6. Suppl Table 1 is really essential to the interpretation of this data and should be a main table.

We have now incorporated this table into the main tables of the manuscript as Table 1 (page 31).

Reviewer #3 (Remarks to the Author):

This is the latest in a series of manuscripts to identify novel loci for sleep-related phenotypes in large samples. While other studies (Dashti et al., BioRxiv 2018, Lane et al. BioRxiv 2018, Jansen et al. BioRxiv 2018) have been based upon self-report rather than actigraphy (accelerometer-derived data), support for the loci identified in two of those studies has been sought in the actigraphy data analysed here.

1. The authors identify 47 genetic associations surpassing the standard genome-wide threshold of significance ($P < 5 \times 10^{-8}$). In the replication sample, direction of effect is convincingly replicated for the top 20 associations with $P < 8 \times 10^{-10}$, but much less so for the remaining 27. When reporting on individual loci, it would be useful to indicate to which of these two groups they belong.

We have updated the results section of the manuscript to present the association P-value for the respective trait as required to make it clearer which group each locus corresponds to.

2. A brief mention of the potential for ascertainment/survival bias would be useful when first presenting the APOE association, rather than leaving it to the final discussion (the description of this in the discussion section could also be a bit clearer).

We have updated the results section containing the APOE finding to mention the possibility of ascertainment bias in the UK Biobank as follows (page 8):

“One possible explanation for this finding is ascertainment bias in the UK Biobank whereby carriers of $\epsilon 4$ risk allele are protected from cognitive decline through other factors.”

We have also modified the discussion to clarify this point as follows (page 14-15):

“Older UK Biobank participants, with the highest risk of cognitive decline with an $\epsilon 4/\epsilon 4$ haplotype and agreeing to an accelerometer-based experiment could be protected from cognitive decline because of selection bias due to other factors⁵⁷. To participate in the UK Biobank study and agree to accelerometer data collection several years after study baseline is less likely to occur in individuals who are in cognitive decline. As a result, the $\epsilon 4$ risk allele may present an association with higher sleep quality. Consistent with this potential bias, the $\epsilon 4$ allele association with reduced numbers of nocturnal sleep episodes is stronger in older age. For example, when splitting individuals by median age, the per allele effect on number of sleep episodes was twice that of the older versus younger group.”

3. The conclusion that lower genetic correlation between self-report and actigraphy-measured sleep duration ‘... suggests differences in the genetic contribution to variation in self-reported versus objective sleep duration’ seems a bit strange. Surely both measures seek to capture the same trait, the difference being that actigraphy is presumably the more accurate of the two. Both actigraphy and self-report measures of sleep duration appear to have been analysed here on the same set of ~85,670 individuals (the methods are not completely clear on this point). Is the higher correlation between actigraphy-measured sleep and activity timings and self-report chronotype possibly due to chronotype being analysed in a much larger study of 449,734 individuals? If self-report chronotype is analysed in the same set of individuals as the actigraphy

data, do self-report & objective measures still show much greater correlation than for sleep duration?

While intuitively it may seem that we are trying to capture the same trait (sleep duration), the subjective nature of self-reporting will mean that some of the genetic associations may be capturing self-perception linked to overall well-being. Therefore, some of the associations may be linked to such features.

The genetic correlations reported were based on comparing the GWAS results in ~85,670 individuals in UK Biobank against the published results of self-report sleep-duration and chronotype measures from Jones et al 2016, PLoS Genetics in 128,266 (pubmed ID: 27494321 (Supplementary Table 10 (previously Supplementary Table 11))). Given the same published study was used with similar numbers for both chronotype (N=127,898) and sleep duration (N = 127,573), we believe the higher genetic correlation between activity-timing and chronotype is unlikely be owing to differences in sample size.

4. The description of the gene-set enrichment analysis performed using MAGMA is minimal. How was Bonferroni correction performed – separately for each trait? How many gene sets were tested (& from which sources)? What settings were used in MAGMA – was a window used to assign nearby SNPs to genes; which gene analysis model was used; was the gene-set test competitive or self-contained...? This information should be presented in the methods. Despite all sleep traits presumably being tested, only one finding is reported (enrichment of serotonin metabolism genes for association with number of nocturnal sleep episodes). Is this the only enrichment that survives correction for this trait? Are there no associations for any other trait? All associations surviving correction for each trait (if any) should at least be presented in a supplementary table.

We thank the reviewer for highlighting these issues. With respect to the detail about the MAGMA analyses, information on pathway databases and Bonferroni thresholds are provided in the cited article (reference 44). However, we do acknowledge that more detail could be provided and have updated the “Gene-set, tissue expression enrichment, and overlap with GWAS-catalog analyses” subsection of the Methods section to include details of the settings we used as follows (pages 20-21):

“For lead and candidate SNP identification, the default settings were used: lead variants were required to have a minimum P-value of 5×10^{-8} ; r^2 threshold for defining LD structure of lead variants was set to 0.6; the maximum P-value cut-off was set to 0.05; the reference panel population was chosen to be 1000 Genomes Phase 3; variants in the reference panel but not in the GWAS were included; the minimum minor allele frequency was set to 0.01 and a maximum allowed distance of 250k between LD blocks was used for variants to be included in the same locus. For assigning variants to genes for gene-set enrichment analyses, positional mapping was used with variants assigned to a gene if they were within the gene start and end points (by setting the distance either side to 0kb) and only protein-coding genes were included in the mapping process. MAGMA gene-set enrichment analysis, implemented in FUMA, adopts a competitive test of gene-set enrichment using 10,894 gene sets obtained from MSigDB⁶⁸. Tested gene sets include BioCarta, REACTOME, KEGG and GO amongst others; a full list of gene sets used by MSigDB can be found at <http://software.broadinstitute.org/gsea/msigdb/collections.jsp>.

Bonferroni correction for gene-set enrichment adjusted for the number of gene sets tested and was applied to each phenotype separately.”

In relation to the results we present in Supplementary Table 8, these are the only results to have reached the Bonferroni-corrected threshold and no results were significant after correction for the other phenotypes.

Reviewer #1 (Remarks to the Author):

Thank you for thoroughly and adequately addressing my comments.

I only have one very minor comments:

In the section “Waist-hip-ratio (adjusted for BMI) and educational attainment causally influence sleep outcomes”:

“However, given the genetic correlation and MR analyses are not independent”: I think that the multiple testing correction is not related (or at least shouldn't be related) to the lack of independence between the genetic correlation and MR analysis, so I recommend deleting this sentence.

Reviewer #2 (Remarks to the Author):

Thank you for considering my suggestions re: accelerometer-derived data, immobility, RLS and PLMs. I believe the revised manuscript has not taken my reservations seriously enough. Further look at the literature cited shows that the investigator own data support that these actigraphy measures behave differently in subjects with sleep disorders. Below I am now suggesting examples of improved language if a further revised manuscript is to be considered. It has to be stronger.

Abstract:

“8 accelerometer-derived sleep traits” could be: “Sleep trait estimated through accelerometer studies”

“Although accelerometer-derived measures of sleep are imperfect, these findings provide new biological insights into sleep characteristics in comparison to previous efforts based on subjective measures of sleep.” could be:

“Although accelerometer-derived measures of sleep are imperfect and may be affected by restless legs syndrome, these findings provide new biological insights into sleep characteristics in comparison to previous efforts based on subjective (sleep log) measures of sleep.”

Introduction:

“genetic variants associated with objective measures of sleep 119 and rest-activity patterns”

could be “genetic variants associated with accelerometer-derived measures of sleep 119 and rest-activity patterns”

“Additionally, PSG is relatively burdensome for the participant making it less suitable for measuring sleep over multiple nights and capturing inter-daily variability.” could be “Additionally, PSG is relatively burdensome and involves heavy equipment that disturbs sleep making it less suitable for measuring sleep over multiple nights and capturing inter-daily variability.

“By contrast, research-grade activity monitors (accelerometers), also known as actigraphy devices, provide cost-effective estimates of sleep. However, accelerometer-based studies have often involved much smaller sample sizes than those required for GWAS and have generally focussed on daytime activity^{27,28}. “ could be “By contrast, research-grade activity monitors (accelerometers), also known as actigraphy devices, are more objective and may provide cost-effective estimates of sleep for large scale GWAS (limited sample size have been studied,^{27,28}), although these also have limitations. Indeed, measures of sleep using actigraphy are intrinsically difficult to interpret as awake and not moving cannot be distinguished from sleep. Further, although small studies have shown limited effects of sleep apneas, the effect on large scale data is difficult to assess, as is the effect of periodic leg movements during sleep (PLMS). Sleep apnea (more than 15 events per hour) affects about 20% of the population and PLMS (more than 15 movements per hour), a trait associated with RLS but not exclusively is very frequent, for example affecting 40% of the population older than 60 years old. “ (add refence)

“that demonstrated an acceptable level of agreement with polysomnography and sleep diaries^{29,30}.”

Reference 29 reports on only 28 subjects having had PSGs, 19 of which “had a sleep disorder (again making my point exactly): and does not comment at all on the effect of sleep pathologies on activity recordings. It comments “The agreement between accelerometer estimates of sleep using our algorithm and polysomnography derived parameters was good, as shown in Table 3. For example, sleep parameters derived with a 5 minute window and a 5 degree angle threshold had on average a 31 minute overestimation of sleep duration and an 83% accuracy (Table 3).” This is not what I call good agreement. A 30 minute overestimation (my point exactly about no activity=sleep) is 10% of usual sleep time. A 20% variance in all subjects (may be higher in sleep disorder subjects) could have tremendous effects in 80,000 subjects.

Reference 30 is on 22 good sleepers 28 subjects with sleep disorders. These subjects had various sleep disorders: hypersomnia (N = 2), insomnia (N = 2), REM behavior disorder (N = 3), sleep apnea (N = 5), narcolepsy (N = 1), sleep apnea (N = 4), parasomnia (N = 1), restless leg syndrome (N = 5), and

sleep paralysis (N = 1), and nocturnia (N = 1) The behavior of the actigraphy device is clearly different in the two groups (table 4 and 5) and curiously there is no statistics comparing the behavior of the device between the two groups (it would be significant).

Discussion

“Our analysis presents the first large-scale GWAS of multiple sleep traits estimated 388 from accelerometer data using our activity-monitor sleep algorithm, with estimates previously demonstrated to be highly correlated with polysomnography^{29,30}”

Same problem.

Reviewer #3 (Remarks to the Author):

Looks fine. No further comments on the manuscript.

Reviewers' comments:

Reviewer #1 (Remarks to the Author):

Thank you for thoroughly and adequately addressing my comments.

I only have one very minor comments:

In the section “Waist-hip-ratio (adjusted for BMI) and educational attainment causally influence sleep outcomes”:

“However, given the genetic correlation and MR analyses are not independent”: I think that the multiple testing correction is not related (or at least shouldn't be related) to the lack of independent between the genetic correlation and MR analysis, so I recommend deleting this sentence.

We thank the reviewer from their comment. We feel that is important to highlight the fact that these analyses are likely to be correlated. This is because a higher genetic correlation may produce a higher causal effect in at least one direction. Therefore, providing a threshold that reflects the total number of MR analyses that could have been performed is important ($P = 0.05 / (8 \times 234)$).

Reviewer #2 (Remarks to the Author):

Thank you for considering my suggestions re: accelerometer-derived data, immobility, RLS and PLMs. I believe the revised manuscript has not taken my reservations seriously enough. Further look at the literature cited shows that the investigator own data support that these actigraphy measures behave differently in subjects with sleep disorders. Below I am now suggesting examples of improved language if a further revised manuscript is to be considered. It has to be stronger.

We would like to thank the reviewer for providing us with suggested sentences to strengthen the paper.

Abstract:

1. “8 accelerometer-derived sleep traits” could be: “Sleep trait estimated through accelerometer studies”

We have changed the text in the sentence from:

“Using accelerometer data from 85,670 individuals in the UK Biobank, we performed a genome-wide association study of 8 accelerometer-derived sleep traits, 5 of which are not accessible through self-report alone.”

to:

“Using accelerometer data from 85,670 individuals in the UK Biobank, we performed a genome-wide association study of 8 sleep traits estimated using accelerometer data, 5 of which are not accessible through self-report alone.”

2. “Although accelerometer-derived measures of sleep are imperfect, these findings provide new biological insights into sleep characteristics in comparison to previous efforts based on subjective measures of sleep.” could be:

“Although accelerometer-derived measures of sleep are imperfect and may be affected by restless legs syndrome, these findings provide new biological insights into sleep characteristics in comparison to previous efforts based on subjective (sleep log) measures of sleep.”

We have changed the text in the sentence to read:

“Although accelerometer-derived measures of sleep are imperfect and may be affected by restless legs syndrome, these findings provide new biological insights into sleep characteristics in comparison to previous efforts based on subjective measures of sleep.”

3. Introduction:

“genetic variants associated with objective measures of sleep 119 and rest-activity patterns” could be “genetic variants associated with accelerometer-derived measures of sleep 119 and rest-activity patterns”

We have made the change as suggested:

“In this study we identify genetic variants associated with accelerometer-derived measures of sleep and rest-activity patterns and use them to further understand the biology of sleep.”

In addition, we have replaced “objective measures” with “accelerometer-derived” where applicable throughout the main text.

4. “Additionally, PSG is relatively burdensome for the participant making it less suitable for measuring sleep over multiple nights and capturing inter-daily variability.” could be “Additionally, PSG is relatively burdensome and involves heavy equipment that disturbs sleep making it less suitable for measuring sleep over multiple nights and capturing inter-daily variability.

We have changed the text in the sentence to read:

“Additionally, PSG is relatively burdensome, since it involves the use of complex equipment and experimental settings to represent individual’s habitual sleep, making it less suitable for measuring sleep over multiple nights and capturing inter-daily variability.”

5. “By contrast, research-grade activity monitors (accelerometers), also known as actigraphy devices, provide cost-effective estimates of sleep. However, accelerometer-based studies have often involved much smaller sample sizes than those required for GWAS and have generally focussed on daytime activity^{27,28}. “ could be

“By contrast, research-grade activity monitors (accelerometers), also known as actigraphy devices, are more objective and may provide cost-effective estimates of sleep for large scale GWAS (limited sample size have been studied,^{27,28}), although these also have limitations. Indeed, measures of sleep using actigraphy are intrinsically difficult to interpret as awake and not moving cannot be distinguished from sleep. Further, although small studies have shown limited effects of sleep apneas, the effect on large scale data is difficult to assess, as is the effect of periodic leg movements during sleep (PLMS). Sleep apnea (more than 15 events per hour) affects about 20% of the population and PLMS (more than 15 movements per hour), a trait associated with RLS but not exclusively is very frequent, for example affecting 40% of the population older than 60 years old. “ (add reference)

We have changed part of the introduction to read:

“By contrast, research-grade activity monitors (accelerometers), also known as actigraphy devices, are more objective and may provide cost-effective estimates of sleep for large studies. To date, studies of limited sample size have been performed and focussed on daytime activity^{27,28}.”

We have also extended the limitations that form the final paragraph of the discussion section with the following text:

“Measures of sleep using actigraphy are intrinsically difficult to interpret, as awake and not moving cannot be distinguished from sleep. Furthermore, although small

studies have shown limited effects of events that disturb sleep (such as respiratory events or periodic limb movements), the effect on large scale data is difficult to assess. Moderate to severe sleep apnea (≥ 15 apneas or hypopneas per hour of sleep) and periodic limb movements in sleep (≥ 15 movements per hour of sleep) are relatively common in individuals within the age range of the present study^{59,60}. Unfortunately, these conditions were not captured well in the UK Biobank study, limiting the possibility of evaluating the effects of such sleep disorders on accelerometer-derived sleep traits. Future studies of PSG-derived metrics of sleep, such as total sleep duration, sleep efficiency, and proportions of sleep stages should be conducted.”

6. “that demonstrated an acceptable level of agreement with polysomnography and sleep diaries^{29,30}.”

Reference 29 reports on only 28 subjects having had PSGs, 19 of which “had a sleep disorder (again making my point exactly): and does not comment at all on the effect of sleep pathologies on activity recordings. It comments “The agreement between accelerometer estimates of sleep using our algorithm and polysomnography derived parameters was good, as shown in Table 3. For example, sleep parameters derived with a 5 minute window and a 5 degree angle threshold had on average a 31 minute overestimation of sleep duration and an 83% accuracy (Table 3).” This is not what I call good agreement. A 30 minute overestimation (my point exactly about no activity=sleep) is 10% of usual sleep time. A 20% variance in all subjects (may be higher in sleep disorder subjects) could have tremendous effects in 80,000 subjects.

We realize that the term ‘agreement’ might have been used inappropriately in the context of the referenced sentence. The previous study from our group (ref 29) was to demonstrate how accelerometer measurements can be used to estimate sleep. We agree and understand that this measurement is not perfect. However, these differences can be outweighed by the relevance of findings regarding rest-activity regulation provided by this study. Nevertheless, we endorse that future studies with PSG derived metrics of sleep, such as total sleep time, sleep efficiency and proportions of sleep stages should be conducted in the discussion section. Therefore, we have re-worded the text in the highlighted sentence from:

“We used accelerometer data from the UK Biobank to extract estimates of sleep characteristics using a heuristic method previously compared against independent PSG and sleep-diary datasets that demonstrated an acceptable level of agreement with polysomnography and sleep diaries^{29,30}”

To:

“We used accelerometer data from the UK Biobank to extract estimates of sleep characteristics using a heuristic method previously compared against independent PSG and sleep-diary datasets. These estimates have previously been demonstrated to be highly correlated with polysomnography and sleep diaries^{29,30}.”

7. Reference 30 is on 22 good sleepers 28 subjects with sleep disorders. These subjects had various sleep disorders: hypersomnia (N = 2), insomnia (N = 2), REM behavior disorder (N = 3), sleep apnea (N = 5), narcolepsy (N = 1), sleep apnea (N = 4), parasomnia (N = 1), restless leg syndrome (N = 5), and sleep paralysis (N = 1), and nocturnia (N = 1) The behavior of the actigraphy device is clearly different in the two groups (table 4 and 5) and curiously there is no statistics comparing the behavior of the device between the two groups (it would be significant).

In our methodological paper (ref 30) we acknowledged the potential limitations of the method when it is used in individuals with a sleep disorder. Improved differences and mean absolute error among individuals without sleep disorders were indicated in the large number of descriptive results provided in ref 30.

8. Discussion

“Our analysis presents the first large-scale GWAS of multiple sleep traits estimated 388 from accelerometer data using our activity-monitor sleep algorithm, with estimates previously demonstrated to be highly correlated with polysomnography^{29,30}”

Same problem.

We have updated the introduction as described in (6) above to refer to “correlation” instead of “agreement” and believe correlation is an acceptable term here.

Reviewer #3 (Remarks to the Author):

Looks fine. No further comments on the manuscript.